# AutoToM: Scaling Model-based Mental Inference via Automated Agent Modeling

**Zhining Zhang**[*]
Peking University
zzn_nzz@stu.pku.edu.cn

**Chuanyang Jin**[*†]
Johns Hopkins University
cjin33@jhu.edu

**Mung Yao Jia**[*]
Johns Hopkins University
mjia8@jhu.edu

**Shunchi Zhang**[*]
Johns Hopkins University
szhan256@jhu.edu

**Tianmin Shu**
Johns Hopkins University
tianmin.shu@jhu.edu

## Abstract

Theory of Mind (ToM), the ability to understand people's minds based on their behavior, is key to developing socially intelligent agents. Current approaches to ToM reasoning either rely on prompting Large Language Models (LLMs), which are prone to systematic errors, or use handcrafted, rigid agent models for model-based inference, which are more robust but fail to generalize across domains. In this work, we introduce *AutoToM*, an automated agent modeling method for scalable, robust, and interpretable mental inference. Given a ToM problem, *AutoToM* first proposes an initial agent model and then performs automated Bayesian inverse planning based on this model, leveraging an LLM backend. Guided by inference uncertainty, it iteratively refines the model by introducing additional mental variables and/or incorporating more timesteps in the context. Across five diverse benchmarks, *AutoToM* outperforms existing ToM methods and even large reasoning models. Additionally, we show that *AutoToM* can produce human-like confidence estimates and enable online mental inference for embodied decision-making.

Links: Project Page | Code

## 1 Introduction

To successfully engage in rich and complex social interactions such as cooperation, communication, and social learning, humans must adequately understand one another's mental states (e.g., goals, beliefs, desires). This ability is termed Theory of Mind (ToM) [49]. Prior works have demonstrated that like human interactions, Theory of Mind is also crucial for the success of human-AI interactions [7, 14, 28]. To safely and productively interact with humans in an open-ended manner, AI systems need to interpret humans' mental states from observed human behavior [5, 45, 44, 31, 33, 53, 51, 21].

There are two primary approaches to developing machine Theory of Mind in recent works. First, with the rapid progress of large language models (LLMs), there has been an increasing interest in directly applying LLMs to reason about people's mental states with prompting strategies such as perspective-taking [48, 37, 22], change-tracking [18], and temporal-spatial reasoning [17]. However, even with these advanced prompting techniques, state-of-the-art LLMs still make systematic errors in complex scenarios [20]. Second, cognitive studies have demonstrated that model-based inference, in particular, Bayesian inverse planning (BIP), can reverse engineer human-like theory of Mind reasoning [4, 43, 3, 52]. BIP relies on Bayesian Theory of Mind (BToM) models [3] to approximate rational agent behaviors. Inspired by this, recent works have proposed to combine BIP and LLMs to

---

[*]Equal contribution.
[†]Project Lead.

39th Conference on Neural Information Processing Systems (NeurIPS 2025).

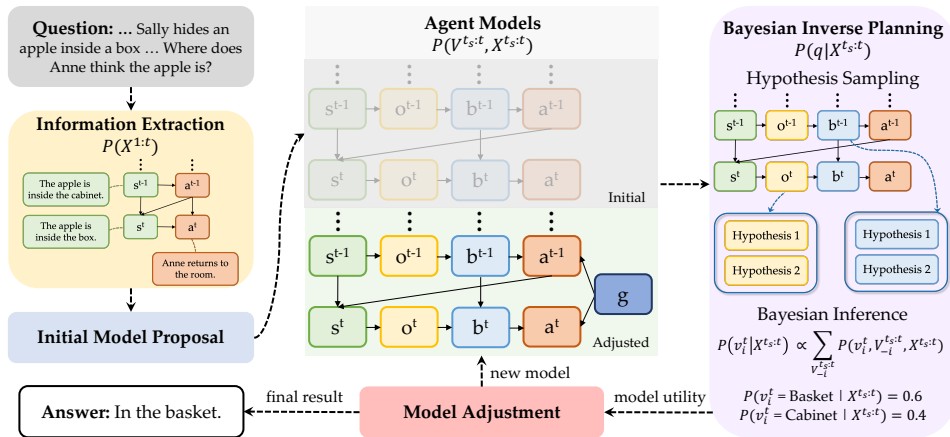

Figure 1: An overview of *AutoToM*. $X^{t_s:t}$ are observable variables, $V^{t_s:t}$ are latent mental variables, and $q$ is the query (in this case, a mental variable $v_i^t \in V^t$). $t_s : t$ denotes timesteps from $t_s$ to $t$ in the context that are considered for inference. Variables $s^t, o^t, b^t, a^t, g^t$ represent state, observation, belief, action, and goal, respectively, with solid arrows indicating dependencies defined in the models. Given a question, we extract the observable variables (information extraction) and propose an initial agent model. This is followed by automated Bayesian inverse planning and iterative model adjustment. When the model utility is high enough, we will produce the final answer based on the inference result.

achieve scalable yet robust model-based ToM inference [20, 39]. While these methods significantly outperform LLMs in specific domains, they typically require manual specification of agent models, including necessary mental variables (e.g., goals, beliefs) for answering a given ToM question. Therefore, they lack the required generalizability for open-ended Theory of Mind.

In this work, we aim to develop a fully *automated* model-based Theory of Mind method. That is a unified method that can be applied to robustly infer any given mental variable in any domain. Achieving this aim requires addressing two critical questions: (1) How can we ensure that our approach is flexible enough to adapt across contexts, robust enough to model diverse human behaviors, and scalable enough to tackle increasingly complex scenarios? (2) How can we avoid manual model specifications and instead automate agent modeling for model-based mental inference?

To address these challenges, we introduce *AutoToM*, a general framework for model-based Theory of Mind. It automates every aspect of Bayesian inverse planning, including the proposal and adjustment of model structures, the identification of relevant timesteps, the generation of hypotheses, and the execution of Bayesian inference. It is designed to operate in *any context*, infer *any mental state*, reason about *any number of agents*, and support *any order of recursive reasoning*, which represents our vision of an open-ended and robust machine Theory of Mind.

Figure 1 provides an overview of *AutoToM*, which consists of two main components: First, **Automated Bayesian Inverse Planning** conducts Bayesian inference based on any given agent model (in the form of a Bayesian network) using an LLM as a computational backend. Unlike prior works that leverages LLMs for Bayesian inverse planning, it has no assumptions about model structure or variable representations. Second, **Automated Agent Model Discovery** iteratively constructs and adjusts an agent model most suitable a given ToM inference problem, eliminating the need for manual model specifications typically required by prior works on model-based ToM inference.

Our main contributions include: (1) a unified formulation of model-based ToM inference; (2) the first approach of automated agent model discovery, AutoToM, for scalable model-based ToM; and (3) a systematic evaluation of AutoToM on multiple ToM benchmarks, cognitive studies, and embodied assistance tasks. The results show that *AutoToM* outperforms state-of-the-art LLMs and large reasoning models, establishing a scalable, robust, and interpretable framework for machine ToM.

## 2   Related Works

**Enhancing LLMs' Theory of Mind.** While LLMs remain limited in achieving robust Theory of Mind inference [42, 38, 10], recent studies have introduced various prompting techniques to enhance

this ability: SimToM [48] encourages LLMs to adopt perspective-taking, PercepToM [22] improves perception-to-belief inference by extracting relevant contextual information, and Huang et al. [18] employ an LLM as a world model to track environmental changes and refine prompts. Explicit symbolic frameworks also contribute: TimeToM [17] constructs a temporal reasoning framework to support inference, SymbolicToM [37] uses graphical representations to track characters' beliefs, and thought-tracing [24] traces multiple hypotheses over time. However, these approaches still exhibit systematic errors in handling long contexts, complex behaviors, and recursive reasoning scenarios.

Among these works, thought-tracing is closely related to ours, as it also maintains hypotheses of mental variables. Compared to thought-tracing [24], *AutoToM* performs explicit agent modeling: it constructs Bayesian networks over mental variables and their causal dependencies, rather than tracking only the queried mental variables. This yields higher robustness to wording or superficial story changes (e.g., no need for wording changes in *AutoToM*), and improves interpretability, as errors can be analyzed through the model structure. Moreover, *AutoToM* adaptively minimizes inference complexity by expanding models only when beneficial, preventing under-/over-modeling and improving efficiency on tasks with longer contexts , more agents, and deeper recursion. By contrast, thought-tracing reweights hypotheses without adjusting model structure or temporal depth.

**Model-based Theory of Mind inference.** Model-based Theory of Mind inference, particularly Bayesian inverse planning (BIP) [4, 43, 3, 52], explicitly constructs representations of agents' mental states and models how these mental states guide behavior through probabilistic agent models. These methods can reverse engineer human ToM inference in simple domains [e.g., 3, 29, 40]. Recent works combine BIP with LLMs to improve ToM inference in more realistic settings [20, 39]. However, they require manual specification of the agent models as well as rigid, domain-specific implementations of Bayesian inference, limiting their adaptability to open-ended scenarios. To overcome this, we propose *AutoToM*, a method for automated agent modeling and mental inference across diverse domains.

**Automated Modeling with LLMs.** There has been an increasing interest in integrating LLMs with inductive reasoning and probabilistic inference for automated modeling. Piriyakulkij et al. [32] combine LLMs with Sequential Monte Carlo to perform probabilistic inference about underlying rules. Qiu et al. [34] further enhance LLM-based inductive reasoning by iteratively proposing, selecting, and refining textual hypotheses of rules. Li et al. [27] employ LLMs to construct, critique, and refine statistical models represented as probabilistic programs for data modeling. Wang et al. [46] prompt LLMs to generate natural language hypotheses that are then implemented as verifiable programs for inductive reasoning. Hypothetical minds [6] leverage LLMs to propose and evaluate agent strategies for multi-agent planning, but do not specifically infer individual mental variables. Our method also aims to achieve automated modeling with LLMs. Unlike prior works, we propose a novel automated model discovery approach for Bayesian inverse planning, where the objective is to confidently infer any mental variable given any context by constructing a suitable agent model.

# 3 AutoToM

## 3.1 Preliminaries: A Unified Formulation of Model-based ToM

Bayesian Inverse Planning (BIP) is a computational framework for model-based ToM inference [4]. It assumes that the agent acts rationally according to a generative agent model [3], which specifies how internal variables lead to observable actions in a Bayesian network (e.g., the example models on the bottom panels in Figure 2a. Using inverse inference, BIP inverts this generative process to assess what latent mental variables can lead to observed agent behavior. This probabilistic inference reasons about how agents make decisions, serving as a robust solution to ToM challenges.

There have been different instantiations of BIP in prior works [e.g., 4, 43, 30, 19]. Here we formally define BIP in a *unified* manner. We denote the observable variables at time $t$ describing the environment and an agent's behaviors as $X^t = \{x_i^t\}_{i \in N_X}$, where $N_X$ is the set of observable variables and $x_i^t$ is a particular variable (state, action, or utterance) at $t$. We can extract the values of these observable variables from the context provided in a ToM problem. We denote an agent's latent mental variables at time $t$ as $V^t = \{v_i^t\}_{i \in N_V}$, where $N_V$ is the set of mental variables and $v_i^t$ is a particular mental variable (e.g., goal, desire, belief) at $t$. BIP formulates an agent model as a Bayesian network that defines $P(V^t, X^t)$, which indicates how the mental variables drive an agent's behavior. Given

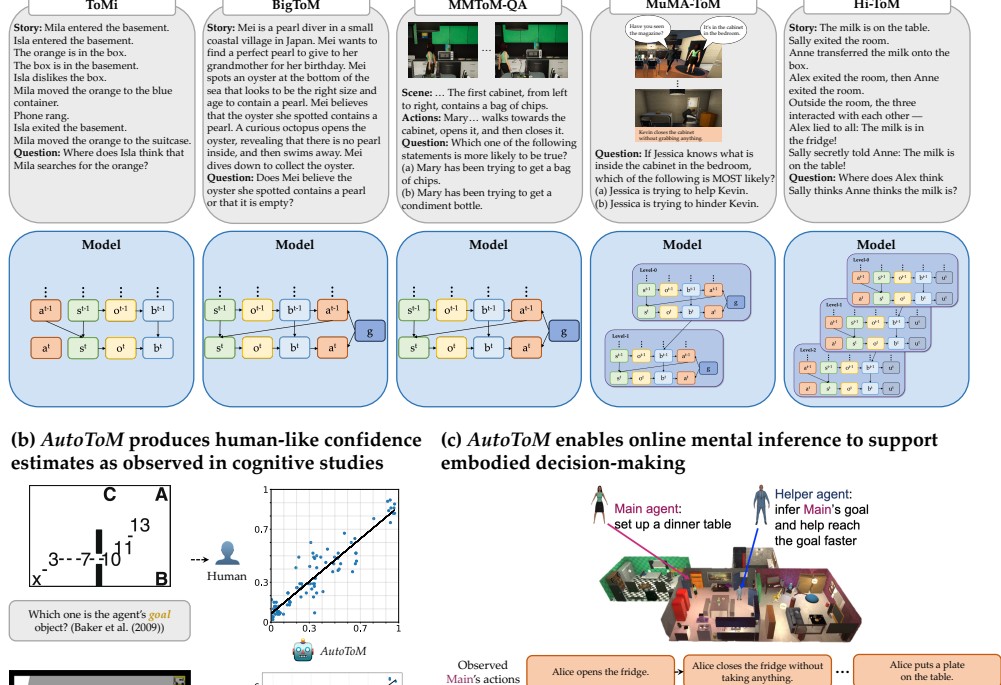

Figure 2: Overview of *AutoToM*'s capacities and applications evaluated in this work. (a) Example questions (top panels) and the necessary agent model for model-based inference (bottom panels) in diverse Theory of Mind benchmarks. Questions in these benchmarks encompass different mental variables, contexts, numbers of agents, the presence or absence of utterances, wording styles, and modalities. (b) *AutoToM* can produce human-like confidence estimation in classic cognitive studies. (c) *AutoToM* can also be used for online goal inference to enhance embodied assistance, where it sequentially updates the inference of a main agent's goal to inform a helper agent's assistance.

this model, BIP infers the latent mental variables for the current step $t$:

$$P(V^t|X^t) = P(V^t, X^t)/\sum_V P(V, X^t) \propto P(V^t, X^t). \tag{1}$$

In many real-world scenarios, past observations (such as actions taken at the previous steps) are often valuable for inferring the mental variables at the current step. Suppose the context from step $t_s$ to step $t$ is relevant for the current mental variable inference, then the inference becomes:

$$P(V^{t_s:t}|X^{t_s:t}) \propto P(V^{t_s:t}, X^{t_s:t}). \tag{2}$$

In a ToM problem, there is a query concerning a specific target variable $q$ to be inferred. We can answer the query via $P(q|X^{t_s:t})$. Typically, the query asks about a latent mental variable $q = v_i^t \in V^t$, the posterior probability is obtained by marginalizing over other latent variables $V_{-i}^{t_s:t}$ which is the subset of $V^{t_s:t}$ excluding $v_i^t$:

$$P(v_i^t|X^{t_s:t}) \propto \sum_{V_{-i}^{t_s:t}} P(v_i^t, V_{-i}^{t_s:t}, X^{t_s:t}). \tag{3}$$

This can also be extended to predicting a future observable variable $q = x_i^{t+1}$:

$$P(x_i^{t+1}|X^{t_s:t}) \propto \sum_{V^{t_s:t}} P(V^{t_s:t}, x_i^{t+1}, X^{t_s:t}). \tag{4}$$

To conduct BIP in different scenarios, we must formulate the mental variables and their causal relationships with agent behavior using suitable agent models. Each model $M$ is uniquely defined by

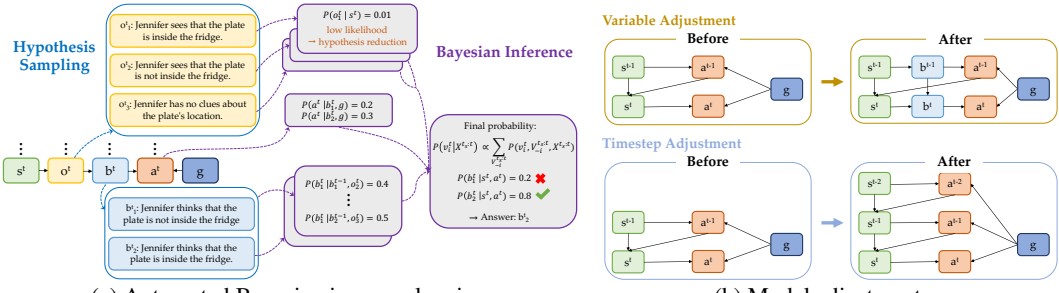

(a) Automated Bayesian inverse planning.        (b) Model adjustments.

Figure 3: (a) Given an agent model, *AutoToM* samples hypotheses for each latent variable ($o^t$ and $b^t$ in this example), remove spurious hypotheses, and conduct Bayesian inference based on estimated local conditionals. (b) Given any ToM inference problem, *AutoToM* refines the agent model by alternating between variable adjustment (introducing belief in this example) and timestep adjustment.

the observable variables and the latent mental variables, i.e., $M = (V^{t_s:t}, X^{t_s:t})$. Let $s^t \in S$ be the state at time $t$, and $a^t \in A$ be the action taken by the agent at time $t$. The current state and action determines the next state $s^{t+1}$. When the agent has an explicit goal $g \in G$, this setup constitutes a Markov Decision Process (MDP). If the agent only has a partial observation of the state, the model becomes a Partially Observable Markov Decision Process (POMDP) [23]. In POMDP, the agent receives a partial observation $o^t$ of the true state $s^t$, maintains a belief $b^t$ over the possible states, and selects its action $a^t$ based on this belief and goal. When there is high-order recursive reasoning between two agents ($i$ and $j$), we can adopt an Interactive POMDP (I-POMDP) [12], where the belief of state at level $l > 0$ for agent $i$ will become the belief of interactive state $is^t = (s, b_{j,l-1}, g_j)$, where $b_{j,l-1}$ is the belief of agent $j$ at the lower level $l - 1$ and $g_j$ is agent $j$'s goal.

### 3.2 Overview of *AutoToM*

As shown in Figure 1, *AutoToM* aims to construct a suitable agent model for Bayesian inverse planning to confidently infer any target variable. There are several key challenges in achieving this: First, different ToM inference problems require different agent models (as illustrated in Figure 2a). Second, our method must determine which timesteps in the context are relevant. Third, there is no predefined hypothesis space for each variable, and each space could be infinite. Last, to infer mental variables in any context, we must flexibly represent them without manual specifications.

*AutoToM* addresses these challenges in the two key components: (1) automated Bayesian inverse planning (Section 3.3), which conducts BIP given a specified agent model, and (2) automated agent model discovery (Section 3.4), which proposes and adjusts the agent model based on the question and the inference results. These two components form a self-improvement loop to iteratively update the agent model and the corresponding inference result. More details are provided in Appendix A.

### 3.3 Automated Bayesian Inverse Planning

Given an agent model, $M$, including the necessary latent mental variables $V^{t_s:t}$ and the observable variables $X^{t_s:t}$, we integrate LLMs as the computational backend to implement every aspect of the Bayesian inverse planning. In particular, the hypothesis sampling module suggests a small set of possible values of latent variables. The Bayesian inference module then computes the posterior distribution of the target variable in the query based on Eqn. (3) or Eqn.( 4).

**Hypothesis Sampling.** Conventional BIP assumes a manually defined hypothesis space and representation for each latent mental variable. Our hypothesis sampling module instead leverages an LLM to propose only a small set of quality hypotheses for each latent variable in $V^{t_s:t}$. This is akin to amortized inference [35, 19]. To ensure that the sampled hypotheses are relevant to the ToM inference, we guide the sampling process with both the question and the observable variables $X^{t_s:t}$. To remove spurious hypotheses generated by the LLM, we further apply *hypothesis reduction* to eliminate unlikely hypotheses and reduce the hypothesis space. Unlikely hypotheses are identified by evaluating the local conditionals. For instance, we discard observation hypotheses with low likelihood conditioned on the state as shown in Figure 3a.

**Bayesian Inference.** As shown in Figure 3a, we estimate each local conditional in $P(V^{t_s:t}, X^{t_s:t})$ using an LLM. After marginalizing the joint distribution over non-target latent variables via explicit calculation, we then produce the posterior probabilities of the target variable, i.e., Eqn. (3). This also applies to predicting a future observable variable, i.e., Eqn. (4).

Our automated BIP greatly generalizes prior methods that combine BIP and LLMs, such as BIP-ALM [20] and LIMP [39]. Specifically, prior methods assume a fixed model structure defined for a specific ToM problem and require handcrafted, domain-specific representations for physical and mental states. They also cannot propose hypotheses for non-target latent variables. For instance, to infer an agent's goal, BIP-ALM conducts a manual belief update while LIMP has no explicit belief update at all. In contrast, *AutoToM* can conduct any ToM inference based on any agent model structure and consider multiple non-target latent variables simultaneously. Additionally, unlike prior methods, our Bayesian inference can work with arbitrary levels of recursion for high-order ToM inference.

### 3.4 Automated Agent Model Discovery

Prior works on model-based ToM inference rely on manually designed agent models, limiting their applicability to domain-specific scenarios. In contrast, the Automated Model Discovery component automatically proposes a model and dynamically adjusts it to ensure both the *effectiveness* of the model—confidently inferring agents' mental states—and the *efficiency* of the inference by minimizing model complexity. To achieve this, we formulate the utility of a model $M = (V^{t_s:t}, X^{t_s:t})$ used for answering a given query $q$ as

$$U(M, q) = R(M, q) - C(M), \tag{5}$$

where $R(M, q)$ assesses the model's confidence in answering the query, and $C(M)$ is its computational cost. In this work, the reward is defined as $R(M, q) = -H(P(q|X^{t_s:t}))$, where $P(q|X^{t_s:t})$ is the probability distribution of the target variable based on Eqn. (3) or Eqn. (4), and $H(\cdot)$ is its entropy. This is designed to decrease the uncertainty in the inference. To minimize the compute needed for the inference, we define the cost of the model as $C(M) = \alpha|M|$, where $|M|$ denotes the model's complexity, measured by the number of latent mental variables, and $\alpha > 0$ is a weighting factor. The cost increases with complexity, encouraging parsimonious models with lower compute.

There are three modules for Automated Model Discovery:

**Information Extraction.** This module extracts the values of observable variables $X^{1:t}$ from the context, including states ($s^t$), actions ($a^t$), and utterances ($u^t$), organized along a timeline (the number of timesteps is determined by the number of actions and utterances). When there are multiple agents, we identify whose mental state the question is asking about (i.e., the target agent), and then construct the timesteps based on the target agent's actions and/or utterances. The extraction is performed once using an LLM and used for model proposal and Bayesian inverse planning.

**Initial Model Proposal.** We employ an LLM to propose an initial agent model based on $X^{1:t}$ and the query. This initial model has minimal complexity, containing only the essential mental variables needed to answer the question. This initial proposal also assesses the level of recursive reasoning necessary for higher-order ToM inference. Note that we always begin with only considering the last timestep in context, i.e., $t_s = t$. Following this model, we conduct automated Bayesian inverse planning, as described in Section 3.3. If the model utility exceeds a threshold $U_{\min}$, we accept the inference result as the final answer. Otherwise, we use the model utility to guide model adjustments.

**Model Adjustment.** We iteratively adjust the proposed model to maximize the utility by considering two types of model adjustments: variable adjustment (Figure 3b) and timestep adjustment (Figure 3b):

*Variable Adjustment.* We refine the model structure at a specific timestep by iteratively introducing new, relevant latent variables into the model to address uncertainty in the inference. These variables include goal, belief, observation, and interactive state as summarized in Table 4 in Appendix A. This follows the typical causal structures introduced in prior decision-making models [e.g., 23, 3, 43, 12]. Such restricted variable adjustment helps reduce the model space and ensures the proposed models can explain human behavior. For each adjustment, we compute the updated model utility and accept the modification that offers the biggest increase in utility. This iterative process continues until no further significant improvements are possible. Note that our method can still propose diverse models beyond standard MDP, POMDP, and I-POMDP, even with this restricted model adjustment. Appendix A.5 provides more details on the model space.

Table 1: Results of all methods on ToM benchmarks, grouped by model types: LLMs, ToM prompting, large reasoning models, and model-based inference. "—" indicates that the domain-specific method is not applicable to the benchmark. The best results are shown in **bold**.

| Method | ToMi | BigToM | MMToM-QA | MuMA-ToM | Hi-ToM | All |
|---|---|---|---|---|---|---|
| Llama 3.1 70B | 72.00 | 77.83 | 43.83 | 55.78 | 35.00 | 56.89 |
| GPT-4o | 77.00 | 82.42 | 44.00 | 63.55 | 50.00 | 63.39 |
| Gemini 2.0 Flash | 66.70 | 82.00 | 48.00 | 55.33 | 52.50 | 60.91 |
| Gemini 2.0 Pro | 71.90 | 86.33 | 50.84 | 62.22 | 57.50 | 65.76 |
| SymbolicToM | **98.60** | — | — | — | 44.50 | — |
| SimToM | 79.90 | 77.50 | 51.00 | 47.63 | 71.00 | 65.41 |
| DeepSeek-R1 | 89.40 | 86.25 | 49.67 | 63.44 | 56.50 | 69.05 |
| Gemini 2.0 Flash Thinking | 78.00 | 82.83 | 54.00 | 82.56 | 73.50 | 74.18 |
| o3-mini-high | 73.10 | **86.92** | 64.67 | 70.00 | **75.00** | 73.94 |
| BIP-ALM | 55.60 | 50.33 | 56.17 | 33.90 | 14.50 | 42.10 |
| LIMP | 44.60 | 61.67 | 55.33 | 76.60 | 6.50 | 48.94 |
| *AutoToM* (w/ GPT-4o) | 88.30 | **86.92** | **83.00** | **81.44** | 72.50 | **82.43** |

*Timestep Adjustment.* If model utility remains low and no significant improvement can be achieved via variable adjustment within the current timesteps $t_s : t$, we incorporate an additional step, $t_s - 1$, to enhance context for inference. Upon adding a timestep, we first apply the initial model structure and then adjust variables accordingly.

We iterate the variable and timestep adjustments until either the model utility exceeds the desired threshold or no further meaningful improvement is possible.

## 4    Experiments

### 4.1    Experiment 1: Evaluation on ToM Benchmarks

**Setting.** We evaluated our method on multiple Theory of Mind benchmarks, including ToMi [26], BigToM [11], MMToM-QA [20], MuMA-ToM [39], and Hi-ToM [15]. The diversity and complexity of these benchmarks pose significant reasoning challenges. For instance, MMToM-QA and MuMA-ToM incorporate both vision and language inputs, while MuMA-ToM and Hi-ToM require higher-order inference. Additionally, MMToM-QA features exceptionally long contexts, and BigToM presents open-ended scenarios.

We compared *AutoToM* against state-of-the-art baselines:

- **LLMs:** Llama 3.1 70B [9], GPT-4o [1], Gemini 2.0 Flash and Gemini 2.0 Pro [41];
- **ToM Prompting for LLMs:** SymbolicToM [37] and SimToM [48];
- **Large Reasoning Models:** DeepSeek-R1 [13], Gemini 2.0 Flash Thinking, and o3-mini-high;
- **Model-based Inference:** BIP-ALM [20] and LIMP [39].

We use GPT-4o as the LLM backend for *AutoToM* and all ToM prompting and model-based inference baselines to ensure a fair comparison. For multimodal benchmarks, MMToM-QA and MuMA-ToM, we adopt the information fusion methods proposed by Jin et al. [20] and Shi et al. [39] to fuse information from visual and text inputs, respectively. The fused information is in text form. We ensure that all methods use the same fused information as their input.

**Results.** The main results are summarized in Table 1. *AutoToM* demonstrates the strongest overall performance among all methods, including large reasoning models. Specifically, it outperforms its LLM backend, GPT-4o, by a large margin. This is because *AutoToM* is more robust for inferring mental states given long contexts with complex environments and agent behavior. It is also more adept at recursive reasoning, which is key to higher-order inference. Compared to prior model-based methods, it exhibits superior generalization across different domains. This is enabled by our agent model discovery and the automated BIP.

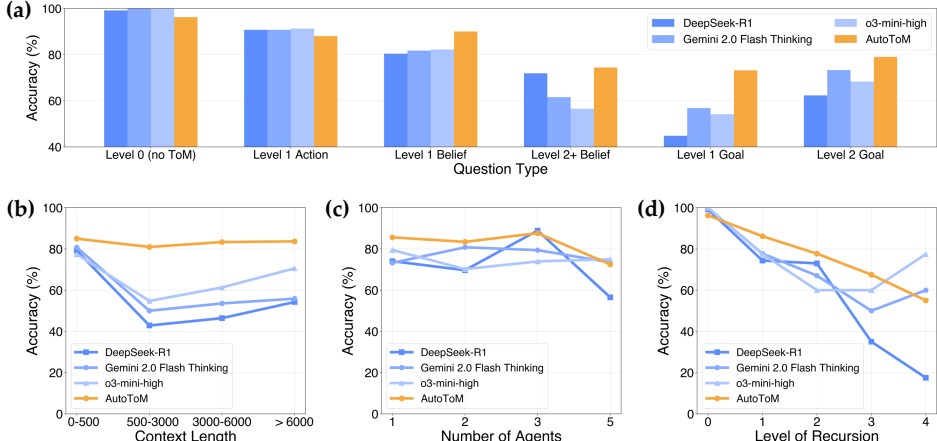

Figure 4: Comparison of *AutoToM* and large reasoning models across various conditions (summarized among all benchmarks): (a) question types, (b) context length, (c) the number of agents, and (d) the level of recursion. Note that "Level 1 Action" refers to Forward Action inference in BigToM, and "Level 2 Goal" refers to the Belief of Goal inference in MuMA-ToM.

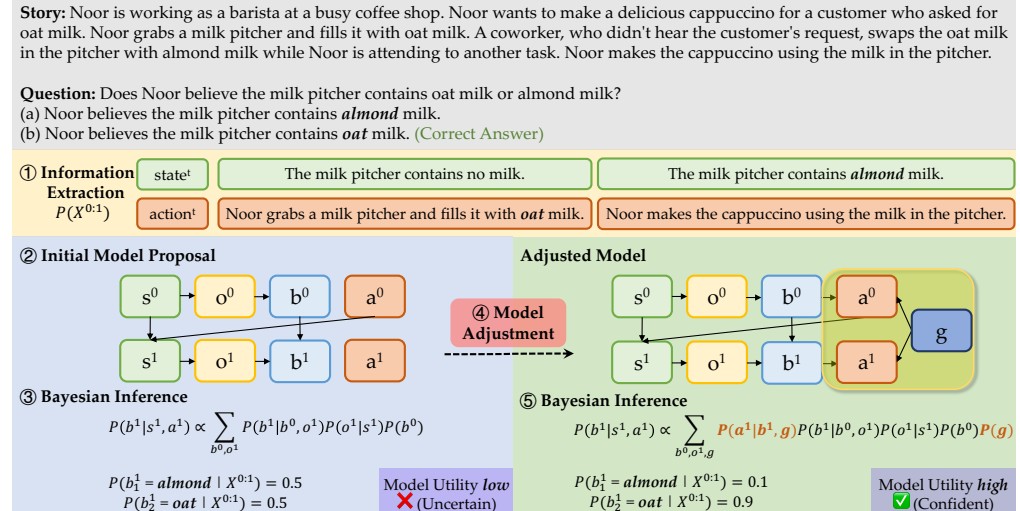

Figure 5: A qualitative example of *AutoToM*'s model adjustment and inference process in a false-belief scenario from BigToM [11]. We show the results from each key model step. It demonstrates how *AutoToM* adjusts the agent model to increase inference confidence. Detailed procedures of Bayesian inference for both the proposed and adjusted models are provided in Appendix C.5.

We also compared the performance of *AutoToM* with large reasoning models across different conditions, summarized over all benchmarks. These include question types, the context length, the number of agents, and the level of recursion. As shown in Figure 4, *AutoToM* demonstrates robust scalability and exhibits a much lower degree of volatility under different conditions than large reasoning models. We provide additional results and evaluations in Appendix C.2 and C.3.

We further report the token cost and inference time comparison on MMToM-QA in Appendix C.1. *AutoToM* achieves higher reasoning performance with comparable or lower computational cost, highlighting its efficiency and scalability.

Figure 5 depicts a qualitative example of how model discovery and adjustment can improve inference for a false-belief question in BigToM. Users can use such interpretable explanations to diagnose and identify sources of model errors, and consequently correct model mistakes. Appendix B shows an example of human feedback improving the model using a user interface developed with *AutoToM*.

**Ablation Study.** We evaluated the following variants of *AutoToM* for an ablation study: no hypothesis reduction (**w/o hypo. reduction**); always using POMDP (**w/ POMDP**); always using the initial model proposal without variable adjustment (**w/o variable adj.**); only considering the last timestep (**w/ last**

Table 2: Performance comparison on MMToM-QA. LLM indicates the model itself; *AutoToM* represents our method with the corresponding model as the backend.

|  | LLM | *AutoToM* |
|---|---|---|
| GPT-4o | 44.0 | **83.0** |
| Qwen3-235b-a22b-2507 | 45.0 | **67.5** |
| DeepSeek-chat-v3-0324 | 34.8 | **71.1** |
| Gemini-2.5-Flash (thinking disabled) | 44.7 | **71.7** |

**timestep**); and considering all timesteps without timestep adjustment (**w/ all timesteps**). The results in Figure 6 show that the full *AutoToM* method constructs a suitable agent model, enabling rich ToM inferences while reducing compute. In particular, key model components, including hypothesis reduction, variable adjustment, and timestep adjustment, optimize efficiency without sacrificing performance. Full ablation results are provided in Appendix C.4.

**Sensitivity to LLM Backends.** To test *AutoToM*'s performance sensitivity to LLM backends, we conducted additional experiments using alternative models. Note that we used the same prompt for each backend LLM. Specifically, we replace the GPT-4o backend with Qwen3-235B (open-sourced), DeepSeek-V3 (open-sourced), and Gemini-2.5-flash (thinking disabled) on the most challenging MMToM-QA benchmark. Notably, *AutoToM* with any LLM as the backend outperforms the corresponding LLM performance by a large margin (Table 2). Crucially, we achieve this without extra prompt engineering.

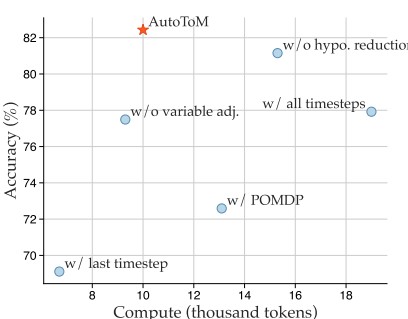

Figure 6: Averaged performance and compute of *AutoToM* (star) and its variants (circles) on all benchmarks.

**Statistical Reliability.** To assess result stability, we additionally ran multiple trials on the most challenging benchmark, MMToM-QA. Across three different random seeds, *AutoToM* achieved a mean accuracy of 82.56% with a standard error of 0.45%, which is consistent with the 83.00% reported in Table 1. Similarly, o3-mini-high achieved a mean accuracy of 65.94% with a standard error of 0.59%. These results indicate that the evaluation is stable across runs, and our conclusions remain robust.

## 4.2 Experiment 2: Evaluation on Classic Cognitive Studies

**Setting.** *AutoToM* produces posterior distributions over the hypothesis space, offering uncertainty estimates. This allows us to compare the model uncertainties with human judgments. We adapted two well-known cognitive studies on human ToM: online goal inference in [4] and desire and belief inferences in the food truck scenarios [3]. As shown in Figure 2b, in each study, participants were shown agent behavior in a 2D gridworld and asked to judge the agent's goal in [4] and desires and beliefs in [3]. A capable model needs to sequentially update multiple hypotheses with varying degrees of confidence that closely resemble human judgment.

In this experiment, we generated captions for the frames in both tasks and evaluated *AutoToM* on all available types of scenarios, using the posterior probabilities from *AutoToM* as its confidence. For baseline, we asked GPT-4o and o3-mini-high to produce confidence scores for each hypothesis in all trials, given the same captions. Implementation details are provided in Appendix D.2.

**Results.** We computed the correlation between model responses and human judgments reported in the original studies. As shown in Table 3, *AutoToM* aligns well with human confidence judgments on all three tasks. In particular, *AutoToM* demonstrates a substantially higher correlation with humans than GPT-4o and o3-mini-high in more complex tasks with a partially observable environment. The results indicate that *AutoToM* is able to produce nuanced confidence estimates that closely mirror human inference patterns in different environments. We provide additional results in Appendix D.1.

## 4.3 Experiment 3: Embodied Assistance

Table 3: Pearson correlation coefficients and $p$-values between model and human judgments. Strong and significant correlations are bolded. $^*$: $p \leq .05$, $^{**}$: $p \leq .001$. "obs." indicates observability.

| Task | AutoToM | GPT-4o | o3-mini-high |
|---|---|---|---|
| Online goal inference (full obs.) in [4] | **0.93**$^{**}$ | **0.81**$^{**}$ | **0.97**$^{**}$ |
| Desire inference (partial obs.) in [3] | **0.88**$^{**}$ | 0.30 | 0.52$^*$ |
| Belief inference (partial obs.) in [3] | **0.73**$^{**}$ | 0.04 | 0.03 |

**Setting.** As recent cognitive studies have suggested, humans routinely utilize ToM to improve our decision making in multi-agent settings [47, 16]. To evaluate whether *AutoToM* can help improve multi-agent decision making, we further evaluated it in an embodied assistance benchmark, Online Watch-And-Help (O-WAH) [33], where a helper agent must simultaneously observe a main agent's actions, infer its goal, and assist it to reach the inferred goal faster in realistic household environments. In these tasks, a ToM model must update its inference of the main agent's goal based on the latest observations in an online manner. Given the goal inference at each step, we adopted the uncertainty-aware helping planner proposed in [33] to generate helping actions accordingly. There are 4 task categories (setting the table, putting groceries in the fridge, preparing a simple meal, washing dishes). We evaluated each method across 20 episodes, with 5 episodes in each task category. To reduce variance, the results are reported as the average over 3 runs per episode.

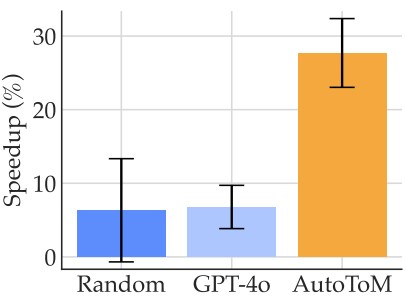

Figure 7: Averaged speedup of *AutoToM* and baselines on the O-WAH benchmark. Error bars indicate standard errors.

As shown in Figure 2c, we applied *AutoToM* to online goal inference. Specifically, *AutoToM* constructs an agent model at each step and maintains the goal hypotheses and corresponding probabilities using Sequential Monte Carlo (SMC) [8]. We also paired the same planner with two baseline goal inference methods: Random Goal (i.e., randomly sampling a goal) and GPT-4o for online goal inference. We did not evaluate any large reasoning models due to their slow inference speed (more than 1 minute per timestep), which makes it impractical for online embodied assistance tasks.

**Results.** As shown in Figure 7, the Random Goal baseline achieves a 6.3% speedup, but with high variance and negative speedup in 50% of the episodes. GPT-4o achieves a similar but more stable speedup of 6.8%. In contrast, *AutoToM* achieves the highest speedup of 27.7%, significantly outperforming all baselines. This is because *AutoToM* can produce more accurate uncertainty estimation of goal hypotheses based on observed actions, which is key to generating robust and useful helping plans. Additional details are provided in Appendix E.

## 5   Conclusion

We have proposed *AutoToM*, a novel framework for scalable model-based Theory of Mind. Given any ToM inference problem, *AutoToM* can automatically construct a suitable agent model and conduct automated Bayesian inverse planning with an LLM backend. Our experimental results have demonstrated that *AutoToM* can answer different Theory of Mind questions in diverse scenarios, significantly outperforming baselines. We have also shown that *AutoToM* can produce human-like confidence estimation about mental inferences in classic cognitive studies, and conduct online goal inference for enhancing embodied assistance in complex household scenarios. *AutoToM* suggests a promising direction toward cognitively grounded ToM modeling that is scalable and robust.

**Limitations and Future Work.** *AutoToM* currently requires a separate process to first fuse information from different modalities into text before inference. In the future, we intend to investigate a natively supported multimodal capacity. Additionally, model adjustments may sometimes fail to recognize the relevance of certain mental variables, resulting in an insufficient model. In the future, we intend to further improve the robustness of *AutoToM* while reducing its inference cost by exploring the possibility of implicit model proposal and Bayesian inference.

## Acknowledgments and Disclosure of Funding

This work was supported by a grant from Amazon. The authors would like to thank Hyokun Yun and Tanya Roosta for their helpful comments.

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

---

**Algorithm 1** *AutoToM*

---

**Require:** Question $Q$, terminate threshold $U_{\min}$
1: ▷ Automated Bayesian inverse planning
2: **function** BIP($M = (V^{t_s:t}, X^{t_s:t}), q$)
3:     **Sample** hypotheses for latent variables $V^{t_s:t}$
4:     **Conduct** Bayesian inference via LLMs to compute $P(q \mid^{t_s:t})$         ▷ Based on Eqn. (3) or Eqn. (4)
5:     **return** $P(q \mid X^{t_s:t})$
6: **end function**
7: ▷ Automated Model Discovery
8: **Extract** query $q$ from $Q$
9: **Extract** observable variables $X^{1:t}$ from $Q$
10: $t_s \leftarrow t$
11: **while** $t_s \geq 1$ **do**
12:     **Propose** initial $V^{t_s}$
13:     $M \leftarrow (V^{t_s:t}, X^{t_s:t})$
14:     $P(q \mid X^{t_s:t}) \leftarrow \text{BIP}(M, q)$
15:     **Compute** the model utility $U(M, q)$
16:     **while** $V^{t_s}$ does not contain all mental variables **do**
17:         $v_{\text{new}}^{t_s} = \arg\max_{v \notin V^{t_s}} U(M + v, q)$         ▷ Based on results from BIP($M + v, q$)
18:         **if** $U(M + v_{\text{new}}^{t_s}, q) > U(M, q)$ **then**
19:             $M \leftarrow M + v_{\text{new}}^{t_s}$
20:             $P(q \mid X^{t_s:t}) \leftarrow \text{BIP}(M, q)$
21:         **else**
22:             **Exit** loop
23:         **end if**
24:     **end while**
25:     **if** $U(M, q) \geq U_{\min}$ **then**
26:         **Exit** loop
27:     **else**
28:         $t_s \leftarrow t_s - 1$
29:     **end if**
30: **end while**
31: **Return** the answer $A \leftarrow \arg\max_q P(q \mid X^{t_s:t})$

---

# A  *AutoToM* Implementation Details

## A.1  Algorithm

We summarize the overall *AutoToM* algorithm in Algorithm 1. Automated Bayesian Inverse Planning (Section 3.3) corresponds to Lines 2–6. Automated Agent Model Discovery (Section 3.4) corresponds to Lines 8–30: Information Extraction in Lines 8–9, Initial Model Proposal in Lines 12–13, and Model Adjustment in Lines 11–30.

## A.2  Automated Bayesian Inverse Planning

**Hypothesis Sampling.** At each timestep, hypotheses for the latent variables are generated using a Large Language Model (LLM) as the backend, guided by the observed variables. Specifically, when the state is not explicitly provided, the LLM acts as a world model, tracking state changes in the story based on the previous state and current actions. For an agent's observation, the LLM is prompted to adopt the perspective of a character, simulating what that character might see, know, or hear in the given environment (e.g., inside a closed room). If no new observation is available at a specific timestep, we neither generate new observations nor update the belief. Additionally, the LLM proposes plausible hypotheses for the agent's belief and goal based on the available information.

**Hypothesis reduction.** We examine all local conditional probabilities involving a single uncertain variable with multiple hypotheses and eliminate those hypotheses that result in significantly low likelihood values. For example, in $P(o^t \mid s^t)$, where $s^t$ represents a determined state, any observation hypothesis that yields a low likelihood for this term is discarded. This approach reduces the computational cost of estimating $P(b^t \mid o^t, b^{t-1})$. Similarly, the same principle is applied to $P(a^t \mid b^t, g^t)$

Table 4: Potential variable adjustments, including introducing goal, belief, observation, and interactive state (for high-order ToM). We show the corresponding local conditionals before and after introducing the new variables.

| New Var. | Before | After |
|---|---|---|
| Goal | $P(a^t \mid s^t)$ $P(a^t \mid b^t)$ $P(a^t)$ $P(a^t)$ | $P(a^t \mid s^t, g)P(g)$ $P(a^t \mid b^t, g)P(g)$ $P(a^t \mid s^t, g)P(g)$ $P(a^t \mid b^t, g)P(g)$ |
| Belief | $P(a^t \mid s^t)$ $P(a^t \mid s^t, g)$ | $P(a^t \mid b^t)P(b^t \mid s^t, b^{t-1})$ $P(a^t \mid b^t, g)P(b^t \mid s^t, b^{t-1})$ |
| Observation | $P(b^t \mid s^t, b^{t-1})$ | $P(b^t \mid o^t, b^{t-1})P(o^t \mid s^t)$ |
| Interactive State | $b(s^t)$ | $b(is^t)$ |

and $P(u^t \mid b^t, g^t)$, where unlikely belief hypotheses are removed to further reduce computational complexity.

## A.3 Automated Agent Model Discovery

During model adjustment, *AutoToM* iteratively adjust the proposed model by considering two types of model adjustments: variable adjustment and timestep adjustment. Table 4 summarizes possible variable adjustments at each timestep.

Given a ToM problem and context, when exploring different models during agent model discovery, *AutoToM* can reuse extracted information, proposed hypotheses about certain mental variables, and local conditionals from previously computed models to avoid redundant computation.

In Algorithm 1, we configure the hyperparameters as follows: $\alpha = 0.02$, $U_{\min} = -0.693$.

## A.4 Recursive Reasoning

Interactive Partially Observable Markov Decision Process (I-POMDP) extends POMDP to multi-agent settings by introducing the concept of interactive states, which include agent models into the state space to capture the recursive reasoning process [12]. We denote $is_{i,l}$ as the interactive state of agent $i$ at level $l$. For two agents $i$ and $j$, where agent $i$ is interacting with agent $j$, the interactive states at each level are defined as:

- **Level 0:** $is_{i,1} = s$
- **Level 1:** $is_{i,1} = (s, b_{j,0}, g_j)$ where $b_{j,0}$ is a distribution over $j$'s interactive state at level 0, $is_{j,0}$
- ...

The framework provides a generative model for agents: given agent $i$'s belief of interactive state $b(is_{i,l})$, its action policy will be $\pi(a_i|is_{i,l}, g_i)$, and its utterance policy will be $\pi(u_i|is_{i,l}, g_i)$.

In our implementation, we sample one possible state based on $b(s)$ at level $l$ to approximate the state at level $l-1$ as imagined by the agent at level $l$. We can recursively apply this process until reaching level 0. Based on the state sampled for level 0, we can then conduct the typical automated BIP based on the model structure at that level. This approach can be conveniently applied to arbitrary levels of recursive reasoning, allowing us to answer higher-order Theory of Mind questions using the same method.

## A.5 Agent Model Space

To apply Bayesian Inverse Planning (BIP) across various scenarios, we define the mental variables and their causal relationships with agent behavior using a family of probablistic agent models. These models accommodate different levels of complexity in how agents behave and reason about their environment.

At each timestep $t$, the observable variables are represented by:

$$X^t = \{x_i^t\}_{i \in N_X}, \text{ where } N_X = \{s^t, a^t, u^t\}$$

Here, the state $s^t$ always appear in $X^t$, while either $a^t$ (action) or $u^t$ (utterance) is included at timestep $t$, depending on whether physical motion or verbal communication is presented. In some cases, $a^t$ is only used to update the state and does not affect the inference of beliefs or goals, while in other scenarios it can be crucial for inferring hidden mental states (e.g., an agent's belief or goal).

The latent variables are denoted by

$$V^t = \{v_i^t\}_{i \in N_V}, \text{ where } N_V = \{o^t, b^t, g^t\}$$

Here, the observation $o^t$ is only included when the agent's belief $b^t$ is part of the model, as it updates $b^t$. The goal $g^t$ is included only if it influences action and is relevant to inference. In cases of higher-order recursive reasoning among multiple agents, the belief over the state $b^t(s^t)$ extends to belief over an interactive state $b^t(is^t)$.

Combining these choices at each timestep yields a model space with 30 possible configurations:

- Action/Utterance: which one is included (2 options).
- Belief/Observation: no belief, belief of state, belief of interactive state, belief of state, or belief of interactive state + observation (5 options).
- Action(Utterance)/Goal: no goal (action(utterance) irrelevant), action(utterance) only, or action(utterance) + goal (3 options).

Over a time interval from $t_s$ to $t$, this scales to $30^{t-t_s+1}$ possible models.

**Examples.** In addition to the Markov Decision Process (MDP), Partially Observable Markov Decision Process (POMDP), and Interactive POMDP (I-POMDP) models introduced in Section 3.1, we present additional examples of models from the BToM model space:

- Observation Update Model: Used in the ToMi benchmark (see Figure 2a), this model focuses on how observations update beliefs. Actions are present but only serve to update states and are irrelevant to the inference questions. This model is well-suited for passive scenarios where the focus is on understanding how hidden states produce observable evidence and how the agent updates its beliefs about the world.
- POMDP Variant without Goal: A partially observable scenario in which goals are trivial or irrelevant. This variant emphasizes how partial observability affects belief formation and action selection, without explicit goal-driven behavior.

# B  *AutoToM*: Model Improvement from Human Feedback

*AutoToM* provides strong interpretability and can improve with human feedback. We built a debugging tool, a simplified version displayed in Figure 8, that shows an example of incorporating human-in-the-loop feedback with *AutoToM*. For a given question, the interactive interface provides clear reasoning justifying its choice. The model lists the mental state variables and actions of agents, which were extracted or sampled with the highest probability. Using this information and the highest calculated probabilities, the model explains its reasoning. After the user understands *AutoToM*'s reasoning, they can identify potential faulty reasoning and provide feedback. Providing human feedback can help improve model reasoning.

In the example BigToM problem in Figure 8, the model initially extracts the wrong mental state variables for Kofi. The user can easily identify this error from the model explanation and give feedback. The user reflects on the model about the lack of details needed for Kofi's goal and Kofi's incorrect observation. *AutoToM* can use this updated feedback to clarify essential information, update its reasoning, and improve its accuracy.

> Kofi is a chef at a popular Ghanaian restaurant. Kofi wants to prepare a traditional groundnut soup for the dinner menu. Kofi adds a small container of peanut butter to the soup pot on the stove. An apprentice chef mistakenly thinking it was a seasoning paste removes the peanut butter from the soup and replaces it with tomato paste. Kofi continues cooking the soup.

**Strong Interpretability**

👤 : Does Kofi believe the soup contains peanut butter or tomato paste?

🤖 : Kofi believes that the soup contains tomato paste. ❌

👤 : Explain why.

🤖 : Based on *Kofi's observation of tomato paste in the soup*, it is highly probable that Kofi believes the soup has tomato paste. *Kofi's action of continuing to cook the soup* aligns with this and *Kofi's goal of preparing a traditional groundnut*.

**Human-in-the-loop Feedback**

👤 : We don't know if Kofi sees tomato paste. Kofi's goal lacks the needed detail that he wants peanut butter in the soup.

🤖 : Recomputing with human feedback → Kofi believes the soup contains peanut butter. ✅

Despite Kofi's undetermined observation, *Kofi continuing cooking the soup* aligns strongly with his goal that *he wants to prepare a traditional groundnut soup with peanut butter* and *belief that the soup contains peanut butter*.

Figure 8: A debugging platform showcasing AutoToM's interpretable explanations for its model choice and learning from human feedback to correct its decision for a sample BigToM backward belief problem.

Table 5: Token cost and inference time comparison on MMToM-QA (lower is better). "K" denotes thousands of tokens, and "s" denotes seconds.

| Model | Avg. #Tokens per Question (K) | Avg. Inference Time (s) |
|---|---|---|
| *AutoToM* | **8.0** | 8.5 |
| o3-mini-high | 10.9 | 21.6 |
| Gemini 2.0 Flash Thinking | 8.8 | **6.1** |

## C  More Results and Implementation Details for Experiment 1

### C.1  Token Cost and Inference Time Comparison

We evaluate the computational efficiency of *AutoToM* compared to large reasoning models in terms of token cost and inference time. Table 5 reports the average number of consumed tokens per question and the average inference time on the MMToM-QA benchmark, which is computationally demanding due to its long contexts. Results show that *AutoToM* achieves substantially higher reasoning performance with comparable or lower computational cost.

### C.2  Per-type Accuracy on All Benchmarks

In Tables 6 - 10, we present the results of *AutoToM* and baselines on each question type of all benchmarks. Here we compare general methods that can be applied to all benchmarks.

### C.3  Additional Benchmarks

We evaluated *AutoToM* on additional benchmarks, FANToM [25] for its challenging scenarios and OpenToM [50] for its *affective* Theory of Mind questions.

#### C.3.1  Evaluations on FANToM

To further demonstrate *AutoToM* 's ability to solve false-belief tasks in more complex scenarios, we tested *AutoToM* on FANToM. We randomly selected a subset of 200 false-belief first-order questions with short contexts due to budget constraints.

Table 6: Detailed accuracy for ToMi.

| Question Type | First order | Second order | Reality | Memory | All |
|---|---|---|---|---|---|
| Llama 3.1 70B | 73.75 | 56.25 | 100.00 | 100.00 | 72.00 |
| GPT-4o | 80.25 | 62.25 | 100.00 | 100.00 | 77.00 |
| Gemini 2.0 Flash | 58.50 | 58.25 | 100.00 | 100.00 | 66.70 |
| Gemini 2.0 Pro | 75.00 | 54.75 | 100.00 | 100.00 | 71.90 |
| SymbolicToM | 98.75 | 98.25 | 100.00 | 98.00 | 98.60 |
| SimToM | 84.75 | 65.00 | 100.00 | 100.00 | 79.90 |
| DeepSeek-R1 | 90.75 | 82.75 | 100.00 | 100.00 | 89.40 |
| Gemini 2.0 Flash Thinking | 83.25 | 61.75 | 100.00 | 100.00 | 78.00 |
| o3-mini-high | 79.50 | 53.25 | 100.00 | 100.00 | 73.10 |
| BIP-ALM | 58.00 | 56.25 | 56.00 | 43.00 | 55.60 |
| LIMP | 43.50 | 44.50 | 44.00 | 50.00 | 44.60 |
| *AutoToM* (w/ GPT-4o) | 95.00 | 77.50 | 93.00 | 100.00 | 88.30 |

Table 7: Detailed accuracy for BigToM.

| Question Type | Forward TB | Forward FB | Backward TB | Backward FB | All |
|---|---|---|---|---|---|
| Llama 3.1 70B | 93.75 | 81.00 | 57.00 | 60.50 | 77.83 |
| GPT-4o | 96.00 | 88.50 | 63.50 | 62.00 | 82.42 |
| Gemini 2.0 Flash | 94.25 | 87.50 | 77.50 | 51.00 | 82.00 |
| Gemini 2.0 Pro | 96.00 | 93.75 | 70.00 | 68.50 | 86.33 |
| SimToM | 92.50 | 90.00 | 25.00 | 75.00 | 77.50 |
| DeepSeek-R1 | 89.75 | 90.50 | 74.50 | 82.50 | 86.25 |
| Gemini 2.0 Flash Thinking | 94.75 | 91.50 | 77.50 | 47.00 | 82.83 |
| o3-mini-high | 93.25 | 90.75 | 78.50 | 75.00 | 86.92 |
| BIP-ALM | 71.75 | 32.50 | 69.50 | 24.00 | 50.33 |
| LIMP | 40.75 | 77.75 | 43.00 | 90.00 | 61.67 |
| *AutoToM* (w/ GPT-4o) | 91.25 | 93.75 | 73.00 | 78.50 | 86.92 |

Table 8: Detailed accuracy for MMToM-QA.

| Question Type | Belief | Goal | All |
|---|---|---|---|
| Llama 3.1 70B | 51.33 | 36.33 | 43.83 |
| GPT-4o | 55.67 | 32.33 | 44.00 |
| Gemini 2.0 Flash | 62.67 | 33.33 | 48.00 |
| Gemini 2.0 Pro | 57.00 | 44.67 | 50.84 |
| SimToM | 75.67 | 26.33 | 51.00 |
| DeepSeek-R1 | 63.00 | 36.33 | 49.67 |
| Gemini 2.0 Flash Thinking | 73.33 | 34.67 | 54.00 |
| o3-mini-high | 88.67 | 40.67 | 64.67 |
| BIP-ALM | 64.33 | 48.00 | 56.17 |
| LIMP | 60.00 | 50.67 | 55.33 |
| *AutoToM* (w/ GPT-4o) | 96.67 | 69.33 | 83.00 |

**Results.** *AutoToM*, with a GPT-4o backend, achieved 72.7%, outperforming the GPT-4o baseline, which achieved 57.5%. *AutoToM*, with a Gemini 2.5 Flash backend, achieved 77.9%, outperforming the Gemini 2.5 Flash baseline, which achieved 38%. With either model as the backend LLM, AutoToM improves upon the original baselines.

**Analysis.** *AutoToM* is able to solve false belief questions by extracting the essential variables. In FANToM, *AutoToM* extracts the state of the conversation (the agents in the conversation, if the main agent is currently in the conversation, and the topics discussed), utterances, and observation of the main agent (depending on whether they are in the conversation or not) to infer belief. In contrast,

Table 9: Detailed accuracy for MuMA-ToM.

| Question Type | Belief | Goal | Belief of Goal | All |
|---|---|---|---|---|
| Llama 3.1 70B | 68.67 | 51.33 | 47.33 | 55.78 |
| GPT-4o | 85.33 | 57.00 | 48.33 | 63.55 |
| Gemini 2.0 Flash | 68.33 | 50.67 | 47.00 | 55.33 |
| Gemini 2.0 Pro | 63.00 | 66.67 | 57.00 | 62.22 |
| SimToM | 54.60 | 43.50 | 44.80 | 47.63 |
| DeepSeek-R1 | 74.67 | 53.33 | 62.33 | 63.44 |
| Gemini 2.0 Flash Thinking | 95.33 | 79.00 | 73.33 | 82.56 |
| o3-mini-high | 74.00 | 67.67 | 68.33 | 70.00 |
| BIP-ALM | 41.20 | 34.10 | 30.60 | 33.90 |
| LIMP | 93.40 | 67.70 | 68.70 | 76.60 |
| *AutoToM* (w/ GPT-4o) | 88.33 | 77.00 | 79.00 | 81.44 |

Table 10: Detailed accuracy for HiToM.

| Question Type | Order 0 | Order 1 | Order 2 | Order 3 | Order 4 | All |
|---|---|---|---|---|---|---|
| Llama 3.1 70B | 65.00 | 47.50 | 22.50 | 20.00 | 20.00 | 35.00 |
| GPT-4o | 92.50 | 65.00 | 40.00 | 27.50 | 25.00 | 50.00 |
| Gemini 2.0 Flash | 95.00 | 70.00 | 50.00 | 27.50 | 20.00 | 52.50 |
| Gemini 2.0 Pro | 100.00 | 62.50 | 50.00 | 37.50 | 37.50 | 57.50 |
| SymbolicToM | 62.50 | 57.50 | 25.00 | 32.50 | 45.00 | 44.50 |
| SimToM | 100.00 | 77.50 | 60.00 | 60.00 | 57.50 | 71.00 |
| DeepSeek-R1 | 95.00 | 80.00 | 55.00 | 35.00 | 17.50 | 56.50 |
| Gemini 2.0 Flash Thinking | 100.00 | 85.00 | 72.50 | 50.00 | 60.00 | 73.50 |
| o3-mini-high | 100.00 | 72.50 | 65.00 | 60.00 | 77.50 | 75.00 |
| BIP-ALM | 10.00 | 17.50 | 10.00 | 20.00 | 15.00 | 14.50 |
| LIMP | 5.00 | 10.00 | 7.50 | 2.50 | 7.50 | 6.50 |
| *AutoToM* (w/ GPT-4o) | 95.00 | 75.00 | 70.00 | 67.50 | 55.00 | 72.50 |

the two baselines struggle to accurately extract and track the agent's observation throughout the conversation.

### C.3.2 Evaluations on Affective Reasoning in OpenToM.

We evaluated *AutoToM*'s affective ToM by extending the causal structure to include attitude and preference (all other components unchanged) and testing on all 596 OpenToM attitude questions.

**Results.** Following OpenToM [50], we used Macro-F1 as the evaluation metric. The random baseline is 0.33. GPT-4o achieved 0.48, while *AutoToM* with GPT-4o backend outperformed it with a score of 0.56. *AutoToM* also approached the performance of the large reasoning model o3-mini-high (0.60), indicating its strong affective reasoning capability.

**Analysis.** Answering the attitude questions does not require inverse planning, since the model can just directly perform forward estimation of attitude based on observed events and preference. This explains why *AutoToM* performed similarly compared to o3-mini-high. This is consistent with results for other question types that do not require inverse planning, such as level 0 (no ToM) and level 1 action questions shown in Figure 4a. However, even in the case where inverse planning is not required, *AutoToM* still scores higher than its backend LLM (GPT-4o). We attribute this to *AutoToM*'s ability to extract and focus on variables that are causally relevant to the task, while filtering out spurious cues by design (see [50], Section 2.5) that may mislead GPT-4o.

### C.4 Full Results of the Ablation Study

Table 11 shows the performance of ablated methods compared to the full *AutoToM* method on all benchmarks.

Table 11: Results of ablated methods compared to the full *AutoToM* method.

| Method | ToMi | BigToM | MMToM-QA | MuMA-ToM | Hi-ToM | All |
|---|---|---|---|---|---|---|
| w/o hypo. reduction | 87.60 | 86.17 | 80.83 | 81.67 | 69.50 | 81.15 |
| w/ POMDP | 76.00 | 86.50 | 82.67 | 50.78 | 67.00 | 72.59 |
| w/o variable adj. | 85.80 | 78.25 | 79.00 | 77.89 | 66.50 | 77.49 |
| w/ last timestep | 68.40 | 77.83 | 76.50 | 78.33 | 44.50 | 69.11 |
| w/ all timesteps | 86.00 | 79.09 | 76.17 | 79.33 | 69.00 | 77.92 |
| *AutoToM* | 88.30 | 86.92 | 83.00 | 81.44 | 72.50 | 82.43 |

Table 12: Comparison of ablated models and the full model on the averaged number of tokens per question (in thousands). Lower is better.

| Method | ToMi | BigToM | MMToM-QA | MuMA-ToM | Hi-ToM | All |
|---|---|---|---|---|---|---|
| w/o hypo. reduction | 15.8 | 6.8 | 8.9 | 24.4 | 20.4 | 15.3 |
| w/ POMDP | 14.9 | 5.5 | 6.2 | 20.0 | 18.8 | 13.1 |
| w/o variable adj. | 8.5 | 6.1 | 8.0 | 14.0 | 10.0 | 9.3 |
| w/ last timestep | 7.8 | 6.1 | 3.9 | 11.6 | 4.0 | 6.7 |
| w/ all timesteps | 14.2 | 7.7 | 44.5 | 16.4 | 12.4 | 19.0 |
| *AutoToM* | 9.8 | 6.5 | 8.0 | 13.6 | 12.0 | 10.0 |

Table 13: Comparison of ablated models and the full model on the averaged number of API calls at inference per question. Lower is better.

| Method | ToMi | BigToM | MMToM-QA | MuMA-ToM | Hi-ToM | All |
|---|---|---|---|---|---|---|
| w/o hypo. reduction | 38.91 | 13.99 | 21.72 | 70.73 | 72.58 | 43.59 |
| w/ POMDP | 36.25 | 8.32 | 12.89 | 42.10 | 51.73 | 30.26 |
| w/o variable adj. | 22.91 | 12.99 | 17.51 | 35.76 | 29.81 | 23.80 |
| w/ last timestep | 21.60 | 12.76 | 7.72 | 28.39 | 9.39 | 15.97 |
| w/ all timesteps | 39.83 | 15.95 | 101.28 | 43.25 | 36.27 | 47.32 |
| *AutoToM* | 32.23 | 13.81 | 17.60 | 35.08 | 36.45 | 27.03 |

In Table 12 and 13, we compare the ablated methods and the full model on the averaged number of tokens per question (in thousands) and the averaged number of API calls at inference per question.

## C.5 Detailed Inferences

Figure 9 shows the detailed procedures of Bayesian inferences for the qualitative example in Figure 5.

## C.6 Qualitative Results

Among general methods, AutoToM achieves state-of-the-art results across all benchmarks. We provide two qualitative examples to illustrate the effect of variable adjustment (example 1) and timestep adjustment (example 2). These examples also demonstrate the interpretability of *AutoToM*, as the constructed model offers us insights into how the method is modeling the agent behavior for the inference.

**Example 1: BigToM (Backward Belief Inference)**

> **Story:** Kavya is a florist in a vibrant Indian market. Kavya wants to create a beautiful bouquet of fresh roses for a customer's anniversary celebration. Kavya sees a batch of roses in her shop that appear to be fresh and vibrant. Unbeknownst to her a mischievous monkey sneaks into the shop and nibbles on the rose petals leaving them damaged and unsuitable for the bouquet. Kavya starts arranging the bouquet using the roses she initially saw.

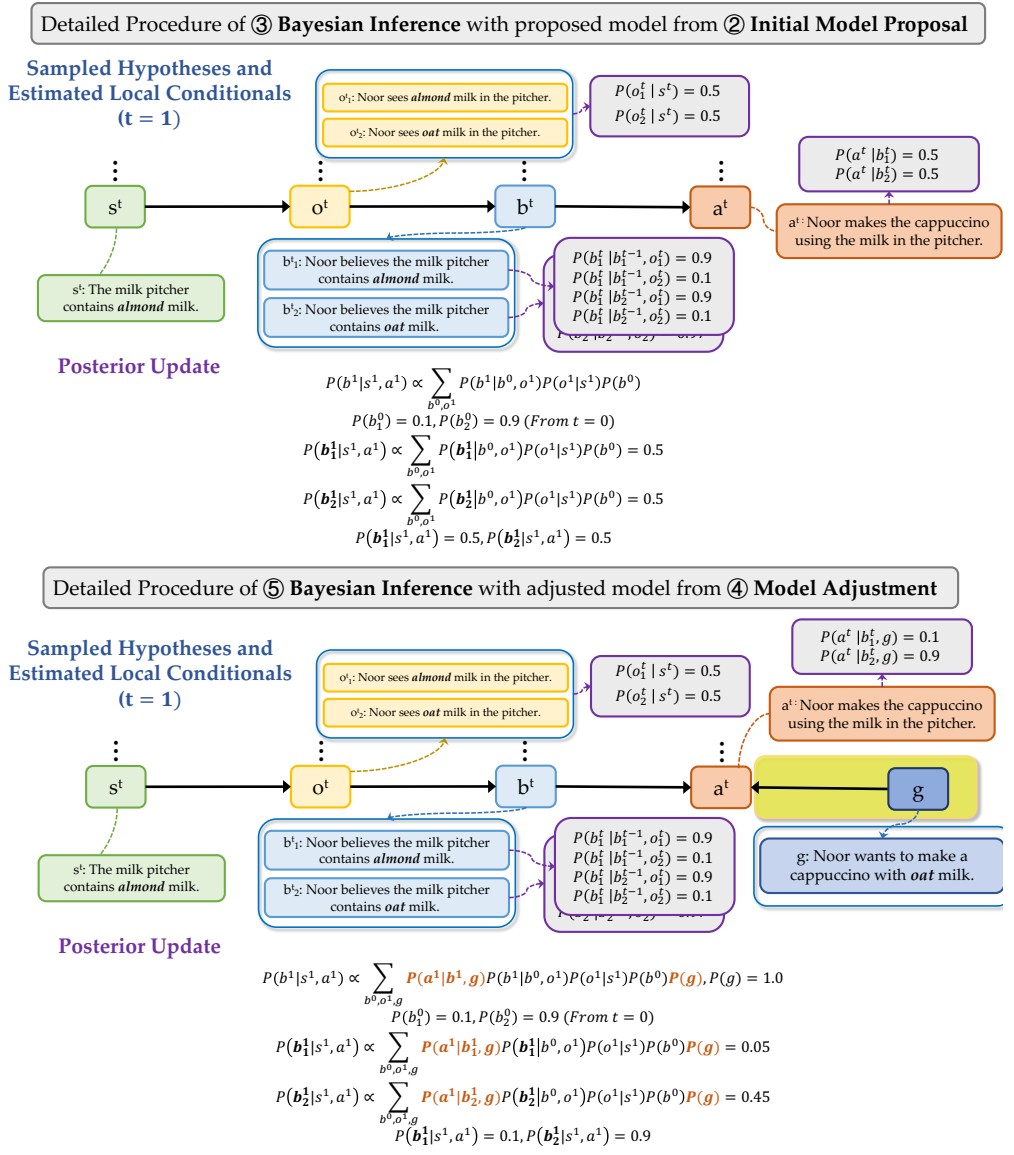

Figure 9: Detailed procedures of how Bayesian inferences are conducted for the proposed and adjusted models in Figure 5. In the initially proposed model, the belief inference results in uncertainty due to ambiguous observations. In the adjusted model, where the agent's goal is explicitly modeled, the use of goal-conditioned action likelihood $P(a^t \mid b^t, g)$ instead of $P(a^t \mid b^t)$ enables more accurate estimation of action likelihoods and leads to improved belief inference with high certainty.

**Question: Does Kavya believe the roses are fresh and perfect for the bouquet or damaged by the monkey?**
(a) Kavya believes the roses are fresh and perfect for the bouquet. **(Correct Answer)**
(b) Kavya believes the roses are damaged by the monkey.

**Variables in the Initial Model Proposal:** State, Observation, Belief

Since the scenario involves only one timestep, a single model suffices. In the initial model, the state of the world indicates that the flowers are damaged after the monkey nibbles on them. However, it remains unclear whether Kavya observes the true condition of the flowers. The model lacks crucial information about Kavya's actions, which are observable and influenced by her beliefs

about the flowers' state. These actions can help infer her true belief. Initially, the probability that Kavya believes the flowers are fresh is moderate, $P$(Kavya believes the roses are fresh and perfect for the bouquet$|X^1) = 0.50$. Without variable adjustment, the model cannot answer the question.

**Variables in the Adjusted Model:** State, Observation, Belief, Action, Goal

For the initial model, the reward is $R(M, q) = -H(P(q|X^{t_s:t})) = -0.693$ and the model cost is $C(M) = \alpha|M| = 0.04$, resulting in a utility $U(M, q) = -0.733$, which does not exceed the utility threshold $U_{\min} = -0.693$. To address the insufficiency of the initial model's utility relative to our termination threshold, we propose an enhanced model incorporating state, observation, belief, action, and goal. In this revised model, Kavya's actions—specifically arranging the bouquet using the roses—align with her goal of creating a beautiful bouquet. These observations allow us to infer with high probability that Kavya believes the roses are fresh and suitable for the bouquet, increasing the belief probability to $P$(Kavya believes the roses are fresh and perfect for the bouquet$|X^1) = 0.97$. With this revised model, the reward is $R(M, q) = -H(P(q|X^{t_s:t})) = -0.135$ and the model cost is $C(M) = \alpha|M| = 0.06$, resulting in a utility $U(M, q) = -0.195$, which exceeds our utility threshold $U_{\min} = -0.693$. Based on the adjusted model, *AutoToM* can confidently determine the correct answer: (a) Kavya believes the roses are fresh and perfect for the bouquet.

**Example 2: MMToM-QA (Belief Inference)**

**Video input:**

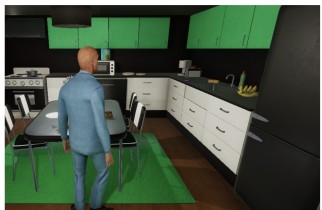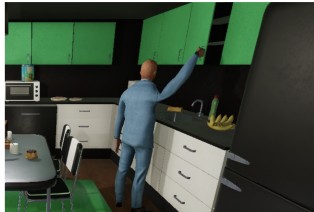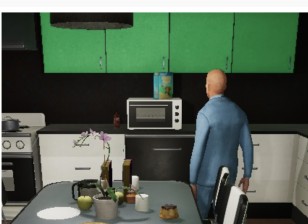

**What's inside the apartment:** The apartment consists of a bedroom, kitchen, living room, and bathroom. In the bedroom, there is a coffee table and a desk. The kitchen is equipped with four cabinets, a fridge, a kitchen table, a microwave, and a stove. The 3rd kitchen cabinet from the left houses a water glass and a dish bowl. Inside the fridge, there are two apples, a salmon, a plate, and a dish bowl. The 2nd kitchen cabinet from the left contains a water glass, a chips, a condiment bottle, and a dish bowl. The 1st kitchen cabinet from the left holds a wineglass, a wine, and a condiment bottle. The microwave contains a salmon, and there is a cupcake in the stove. The 4th kitchen cabinet from the left has a plate. The living room features a cabinet, a sofa, a coffee table, and a desk. Inside the cabinet, there are two apples and four books. A plate and a remote control are placed on the coffee table. The bathroom is furnished with a bathroom cabinet, which is currently empty.

**Actions taken by Mark:** Mark is situated in the bathroom. He proceeds towards the kitchen, making his way to the stove. He opens and then closes the stove. Subsequently, he strides towards the 4th kitchen cabinet, opens it, and then shuts it. He then moves to the 2nd kitchen cabinet, opens and closes it, before doing the same with the 3rd kitchen cabinet. Finally, he heads towards the 1st kitchen cabinet, opens and closes it, and is about to open the microwave.

**Question: If Mark has been trying to get a salmon, which one of the following statements is more likely to be true?**
(a) Mark thinks that the salmon is not inside the microwave.
(b) Mark thinks that the salmon is inside the microwave. **(Correct Answer)**

In this problem, we first fuse the information from text and video following Jin et al. [20]. The fused information is structured into 23 timesteps, each corresponding to an action of Mark at the time. We then propose the initial model: State, Observation, Belief, Action, Goal.

**Without timestep adjustment.** Bayesian inference must be performed sequentially from the first timestep, even though most actions do not contribute to answering the final question. The model

Table 14: Summary of the ToM benchmarks used in the experiments.

| Benchmark | Agent number | Tested concepts | Size | Modality | Communication | Generation | Evaluation |
|---|---|---|---|---|---|---|---|
| ToMi [26] | Multi agents | First & Second Order belief, Reality, Memory | 1000 | Text | No | Templates | Multiple choice Q&A |
| BigToM [11] | Single agent | Belief, Action | 1200 | Text | No | Procedural generation | Q&A |
| MMTOM-QA [20] | Single agent | Belief & Goal | 600 | Text & Video | No | Procedural generation | Multiple choice Q&A |
| MuMA-ToM [39] | Multi agents | Belief, social goal and belief of other's goal | 900 | Text & Video | Yes | Procedural generation | Multiple choice Q&A |
| Hi-ToM [15] | Multi agents | High-order beliefs | 200 | Text | Yes | Procedural Generation | Multiple choice Q&A |

will compute across all timesteps, while the most informative action is actually the last one: if Mark wants to get a salmon but does not believe there is one inside the microwave, he will not open it.

**With timestep adjustment.** We begin inference from the last timestep, where the action likelihood $P(a|b, g)$ is low when $b = $ *Mark thinks that the salmon is not inside the microwave*, and high when $b = $ *Mark thinks that the salmon is inside the microwave*. After performing inference at the last timestep, the belief probabilities corresponding to the choices are 0.998 and 0.002. The reward is given by $R(M, q) = -H(P(q|X^{t_s:t})) = -0.014$, while the model cost is $C(M) = \alpha|M| = 0.06$. This results in a utility of $U(M, q) = -0.074$, which exceeds the threshold $U_{\min} = -0.693$, allowing our model to determine the final answer without considering earlier timesteps.

## C.7   Baseline Implementation Details

For the baselines, we use `gpt-4o-2024-08-06` for GPT-4o, `meta-llama/Llama-3.1-70B-Inst-ruct` from Hugging Face for Llama 3.1 70B, `gemini-2.0-flash` for Gemini 2.0 Flash, `gemini-2.0-pro-exp-02-05` for Gemini 2.0 Pro, `gemini-2.0-flash-thinking-exp-01-21` for Gemini 2.0 Flash Thinking, `o3-mini-2025-01-31` for o3-mini-high, and `deepseek-r1` for Deepseek R1. Among the ToM prompting for LLM benchmarks previously tested on the BigToM dataset, e.g., SimToM, they only tested the subset of the entire dataset with questions for forward action and forward belief and did not test on backward belief questions. With the available SimToM code, we tested it on the full BigToM dataset with GPT-4o.

SymbolicToM maps out the agents' beliefs throughout stories of different levels of reasoning via symbolic graphs. However, the construction of these graphs is specifically designed for the ToMi dataset, where there are fixed actions and sentence formats in the story. Thus it is difficult to generalize to more open-ended scenarios (e.g., BigToM) or stories with multiple agents acting simultaneously (e.g., Hi-ToM). Therefore, we can only evaluate SymbolicToM on ToMi (tested with GPT-4o on the full dataset), for which it was designed.

BIP-ALM and LIMP are both models that combine BIP and LLMs to solve ToM problems. BIP-ALM manually defines symbolic representations of observable and latent variables and assumes POMDP. LIMP is designed to only solve two-level reasoning problems. It uses natural language to represent variables. Both methods assume that the goals are about finding an object and the beliefs are about the locations of that object in a household environment.

## C.8   Benchmark Details

In our evaluation, we test *AutoToM* on BigToM [11], MMToM-QA [20], MuMA-ToM [39], ToMi [26] and Hi-ToM [15]. For ToMi, we use the ToMi dataset that has disambiguated container locations in the story and correctly labeled order of reasoning [2, 36]. For Hi-ToM, we choose the length

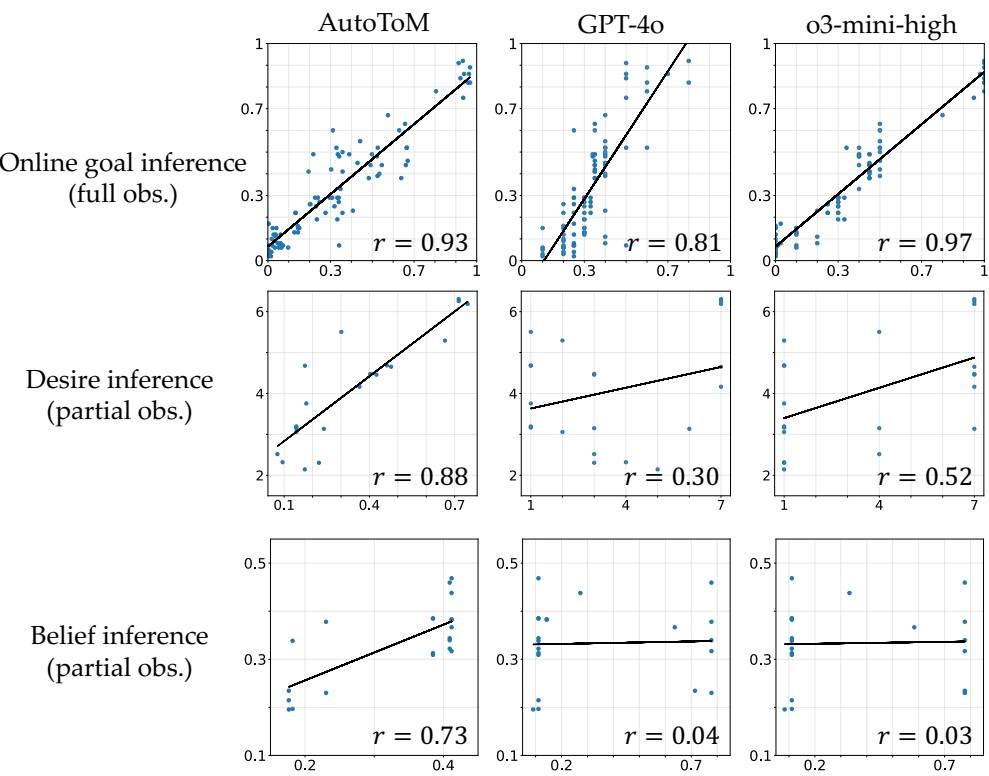

Figure 10: Comparing model and mean human mental state inferences.

1 subset consisting of 200 questions across all orders (0-4) due to the high cost of testing the full dataset.

Table 14 summarizes the benchmarks used to evaluate *AutoToM* against baselines, detailing key features such as test concepts, input modalities, and the number of agents. The results demonstrate that *AutoToM* operates across diverse contexts, infers any mental state, reasons about any number of agents, and supports any level of recursive reasoning.

## D  More Results and Implementation Details for Experiment 2

### D.1  More Results

We provide the scatterplot of human and model judgment fits for all three tasks in Figure 10.

### D.2  Implementation Details

**Scenario Selection and Adaptation.** For the online goal inference task, we selected all 6 usable scenarios (where the human data for each scenario is displayed in the plot) from [4]. For the other two tasks, we adapted from [3], where the original stimuli are grouped into 7 distinct types. We selected one representative scenario from each type, resulting in 7 unique experimental scenarios. This selection was motivated by the fact that scenarios within the same type are highly similar (only minor variations in agent starting positions), which posed challenges for creating clear and distinct natural language narratives. It was also supported by the original study's finding that desire judgments varied minimally within scenarios of the same type.

**Stimuli Translation.** All selected scenarios were translated into natural language descriptions. The generation of the captions was guided by the following principles: (1) We aimed for statements that were complete, clear, and detailed, fully capturing the scenarios and agent trajectories. (2) We focused on describing only what could be objectively observed (physical states, visibilities, and the agent's

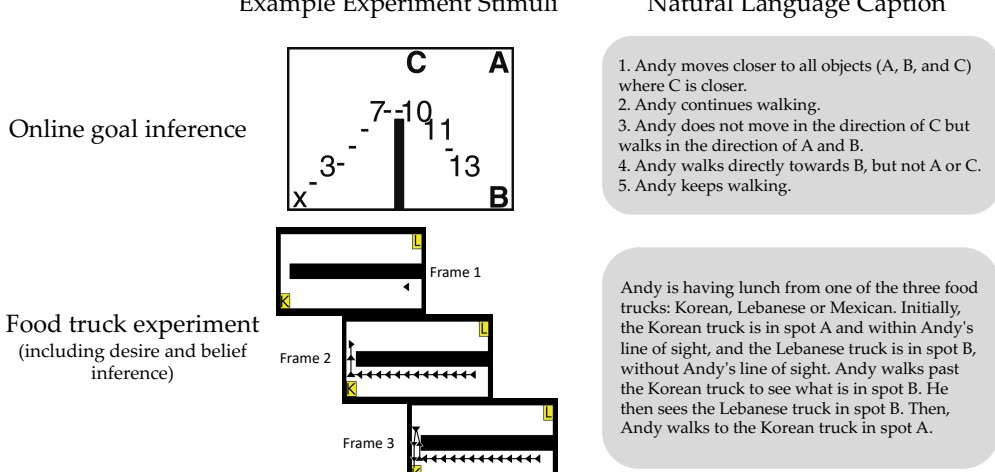

Figure 11: Example inference task scenarios and translated captions in natural language.

actions), without making assumptions about the agent's mental states. Please refer to Figure 11 for caption examples.

**Rationality Assumption.** We incorporated the assumption of approximately rational agents, ensuring consistency with the original studies. This assumption was integrated into the prompts used for estimating action likelihoods in *AutoToM*. To ensure a fair comparison, the identical assumption statement was also included when testing the GPT-4o and o3-mini-high baselines.

**Baseline Evaluation Details.** When testing the baselines, we included the same captions with contexts in the prompt to ask baseline models (1) generate a series of goal probabilities with regard to time steps in task 1 (online goal inference), and (2) provide confidence ratings on a 7-point scale for each belief/goal hypothesis in task 2 (food truck experiment). This process mirrors the judgment task given to human participants in the original experiments.

## E   More Results and Implementation Details for Experiment 3

### E.1   More Results

In the early stages of the episode, GPT-4o may become overly confident in certain hypotheses that lack sufficient cues to inspire confidence. In some cases, such guesses turn out to be correct, resulting in increased speedup. However, more often than not, these hypotheses are incorrect. This can lead to assistance plans that are misaligned with the main agent's goals and cause unnecessary state changes (e.g., moving irrelevant objects), ultimately wasting the helper agent's time and reducing the speedup. In contrast, *AutoToM* recognizes these situations as uncertain and avoids taking actions.

### E.2   Online Mental Inference

The ability to infer and track others' mental states from dynamic cues is key to effective social interactions. We show that *AutoToM* is inherently well-suited for online mental inference by framing it into a sequential Monte Carlo (SMC; [8]) approach that maintains multiple weighted hypotheses about an agent's hidden mental states. We first generate a set of hypotheses (*Sampling*) and assign weights to each hypothesis using *AutoToM* (*Weighting*). As new observations arrive, these weights guide the selection of promising hypotheses and the generation of new hypotheses. The retained hypotheses carry their weights forward, while *AutoToM* evaluates the full past trajectories to assign weights to the new ones (*Propagation*). Based on the latest observation, *AutoToM* then reweights each particle. This particle-tracking method approximates the evolving posterior over mental variables in real time, continually refining both the hypotheses and their likelihood estimates.

### E.3 Implementation Details

As shown in Figure 2c, the online mental inference with *AutoToM* supported the embodied decision-making in Experiment 3, where we used the inferred goal of the main agent at each step to guide the planning of the helper agent. Specifically, the helper began to assist once the most probable goal hypothesis reached a sufficient confidence level (0.8 for picking an object and 0.6 for placing the object at a location).

During online mental inference, we maintained $K = 5$ particles for hypotheses and set a weight threshold of $\tau = 0.1$ for particle filtering. As new observations arrived, promising hypotheses with weights greater than or equal to 0.1 were retained, and new hypotheses were generated to restore a total of 5 particles.

## F   Prompts used in *AutoToM*

### F.1   Information Extraction

We use the following prompts to extract information for each variable in a given question.

---

**Identifying the main agent**

Find the name of the character that we need to infer about in the question and choices. Only output the name. Do not answer the question.

Question: [Question]
Choices: [Choices]
Character name:

---

**Identifying all the agents**

Extract the names of all the characters from the story and question. Provide only the names or roles, without any additional information. Do not answer the question.
Your response should be a list containing the names, like ["name1", "name2"].

Story: [Story]
Response:

---

**Identifying the mental variable to be inferred**

Choose the variable that best summarizes the information about the differences that the choices contain. Only output the variable.

Variables include: [Variables]
Choices: [Choices]
Variable:

---

**Identifying extra information in the question**

If there is any assumed information in the question given (a conditional clause starting with specific words like "if" is contained), rewrite it as a declarative sentence. Do not include any questions in the extra information. Do not make up details for the information. Use the original wording.
Otherwise, output "NONE".

Question: [Question]
Extra Information:

---

## Extracting actions of the main agent

Extract the actions of [Inferred_agent] in the story verbatim without changing any of the original words, pluralizing the words, adding in [Inferred_agent] or any other name, replacing any of the words, replacing pronouns with names or replacing any names with pronouns. Actions of [Inferred_agent] are defined as events that will change the world state, e.g., [Inferred_agent] moving to a new location is an action but [Inferred_agent] being at a location is not an action. If [Inferred_agent] says something, the whole sentence (with "replied", "said") is seen as an action.

Do not change the names of any of the agents, if there is not a name and only a pronoun then just leave the pronoun. There can be more than one agent or more than just the inferred agent. If there are multiple actions in a sentence then they should be extracted as one single action, without changing any of the original words, such as pluralizing the words, replacing any of the words, replacing pronouns with names, or replacing any names with pronouns, and do not add any words.

Do not insert actions, pronouns, or other words that are not explicitly stated in the text. Do not separate the objects in the same action.

Do not add any pronouns. Keep the commas, if any.

Only actions that have already occurred at the time can be considered clearly stated. Again, only extract actions performed by [Inferred_agent].

The output format should be: ["aaa.", "bbb.", ...]. Output only this list.

Story: [Story]
Extraction:

## Extracting actions

Determine if [Character]'s action(s) is clearly stated in the story.

The action(s) cannot be the character's inner thoughts.

Only actions of [Character] that have already occurred, or are currently taking place can be considered clearly stated.

If it's more like [Character]'s desire or goal, it does not count as an action. [Character]'s utterance is considered as an action (include the verb like "said" or "replied" in the evidence sentence, if any). Do not change any of the original wording.

Answer in the form of a list. The first element of the list contains the option A or B. A means clearly stated, and B means not clearly stated.

If the answer is A, include sentence(s) from the original story that serves as evidence, and place it in the second element of the list, without any kind of formatting. Note that there could be multiple action sentences.

Otherwise, the second element can be an empty string. Do not write anything else.

Example 1: ["A", "evidence sentence."]
Example 2: ["B", ""]

Story: [Story]
Answer:

## Extracting beliefs

Determine if the belief of [Character] is clearly stated in the story.

Usually, belief is one's understanding of the state of the world or the state of others. A subjective attitude towards things does not count as belief. An action or utterance of the agent does not count as a belief. Words like "know" or "believe" could be hints for belief.

Answer in the form of a list. The first element of the list contains the option A or B. A means clearly stated, and B means not clearly stated.

If the answer is A, include sentence(s) from the original story that serves as evidence, and place it in the second element of the list, without any kind of formatting.

Otherwise, the second element can be an empty string. Do not write anything else.

Example 1: ["A", "evidence sentence."]
Example 2: ["B", ""]

Story: [Story]
Answer:

## Extracting goals

Determine if the goal of [Character] is clearly stated in the story.
Usually, goals refer to a person's goals or intentions regarding a particular event. Moreover, a sentence that shows a person has been trying to do something, or summarizes their efforts of doing something should always be considered a goal. Helping others to achieve their goals also counts as a person's goal.
Answer in the form of a list. The first element of the list contains the option A or B. A means clearly stated, and B means not clearly stated.
If the answer is A, include sentence(s) from the original story that serves as evidence, and place it in the second element of the list, without any kind of formatting.
Otherwise, the second element can be an empty string. Do not write anything else.
Example 1: ["A", "evidence sentence."]
Example 2: ["B", ""]

Story: [Story]
Answer:

## Extracting observations

Determine if the observation of [Character] is clearly stated in the story.
Observation refers to the main character's perception of an event; it is only considered clearly stated when the protagonist's perception is explicitly mentioned, like if they visually see something, visually notice something, or hear something, or any other state that can be perceived by the agent with but not limited to their 5 senses.
A character's utterance does not mean that their observation is clearly stated, because they might lie.
Answer in the form of a list. The first element of the list contains the option A or B. A means clearly stated, and B means not clearly stated.
If the answer is A, include sentence(s) from the original story that serves as evidence, and place it in the second element of the list, without any kind of formatting.
Otherwise, the second element can be an empty string. Do not write anything else.
Example 1: ["A", "evidence sentence."]
Example 2: ["B", ""]

Story: [Story]
Answer:

## Extracting states

Determine if the story contains the objective state(s) of an object or an event.
State refers to the physical condition of something or the state of the world.
No actions of agents should be involved in the state but it can be the result of an action of an agent. For example, "A entered B" is not a state, while "A is in B" is a state.
An objective state statement should not include personal perspectives but should be objective. If a person's perception is involved, it is no longer considered an objective state.
Answer in the form of a list. The first element of the list contains the option A or B. A means clearly stated, and B means not clearly stated.
If the answer is A, include sentence(s) from the original story that serves as evidence, and place it in the second element of the list, without any kind of formatting.

If there are multiple sentences, include them all in the second element of the list.
Otherwise, the second element can be an empty string. Do not write anything else.
Example 1: ["A", "evidence sentence(s)."]
Example 2: ["B", ""]

Story: [Story]
Answer:

## F.2 Hypothesis Sampling

We use the following prompts to sample hypotheses for the latent variables in the BToM models.

### Sampling beliefs

Propose [num] hypotheses for the belief of [Character] in the story. Usually, belief is one's view or perspective on a state, and it represents an understanding of the physical state of the world. Do not state any reason for the hypotheses. Do not contain any form of explanation in the hypotheses. Output a list of hypotheses of length [num] in following form: ["aaa.", "bbb.", ...]
Given information: [Information] Ensure that the hypotheses align with the given information perfectly. The hypotheses could be like "[Character] believes that A is in B". First, list out all entities in the given information. Then, formulate hypotheses using all entities. Make sure the hypotheses starts with [Character]. Output the hypotheses in the following form: ["aaa."]
Observation Hypotheses: []
Belief Hypotheses:

### Sampling goals

Propose [num] hypotheses for the goal of [Character].
The goal refers to [Character]'s intentions.
Do not provide any explanation for the hypotheses. Do not propose any sentence that's not depicting the goal, like the action or belief of [Character].
The wording for hypotheses cannot be speculative.
The proposed goal does not have to be too specific, e.g., Andy wants to help others; Andy wants to hinder others; Andy is indifferent towards other's goals, etc.
Given information: [Information]
Ensure that the hypotheses align with the given information perfectly. It means that the proposed [Character]'s goal matches what's contained in the information.
Output the hypotheses in the following form: ["aaa."]

Goal Hypotheses: []

### Sampling observations

Propose [num] hypotheses for [Character]'s observation of the world.
The observation refers to [Character]'s current perception of events or the world state. It is only considered clearly stated when [Character]'s perception is explicitly mentioned, like if [Character] sees something or perceives something through other senses. Do not be speculative.
Do not provide any explanation for the hypotheses. Do not propose any sentence that's not depicting the observation, like the action or belief of [Character].
The wording for hypotheses cannot be speculative.
If the information contains "not", make sure the verb for perception (e.g., "see", 'perceives') goes before "not" in the hypotheses. e.g., use 'sees that A is not in B' instead of 'does not see that A is in B' Otherwise, do not include "not" in your hypotheses, and make sure the verb for perception goes first, e.g., 'sees that A is in B'.

Given information: [Information]
Ensure that the hypotheses align with the given information perfectly. It means that when the person has the observation the person will act according to the given information.
First, list all entities in the given information. Then, formulate hypotheses using all entities. Make sure the hypothesis starts with [Character].
Output the hypotheses in the following form: ["aaa."]

Observation Hypotheses: []

## F.3 Likelihood Estimation

We use the following prompts to estimate the likelihood of different variables.

### Estimating the likelihood of the observation given the state

Determine if the statement is likely, and respond with only either A or B.
State: {state}
Here is a statement of {inf_agent}'s current observation. Only evaluate current observation of {inf_agent} based on the state. Do not imagine anything else. Think about {inf_agent}'s location. {inf_agent} is quite likely to observe all objects and events in {inf_agent}'s location, and is unlikely to observe states in another location. If {inf_agent} does not appear in the state, {inf_agent} can't observe anything. Note that the statement has to be precise in wording to be likely. For example, the treasure chest and container are different in wording and they're different objects.

Determine if the following statement is likely: {statement}
A) Likely.
B) Unlikely.

### Estimating the likelihood of the action given the goal and belief and belief of goal

Determine if the statement is likely, and respond with only either A or B.
{inf_agent}'s goal: {goal}
{inf_agent}'s belief: {belief}
{inf_agent}'s belief of other's goal: {belief of goal}
{inf_agent}'s action: {action}
When {inf_agent} wants to help, {inf_agent} is likely to bring an object to other's desired location, and unlikely to grab an object away from other's desired location.
When {inf_agent} wants to hinder, {inf_agent} is likely to grab an object away from other's desired location, and unlikely to bring an object to other's desired location.
When {inf_agent} doesn't know other's goal, {inf_agent} is likely to act according to {inf_agent}'s belief.
If {inf_agent} wants to help and {inf_agent} believes the object is placed at other's desired location, it's unlikely {inf_agent} will move the object.
If {inf_agent}'s goal, {inf_agent}'s belief of goal, and {inf_agent}'s action do not align in any way, the action is unlikely.

Determine if {inf_agent}'s action is likely.
A) Likely.
B) Unlikely.

### Estimating the likelihood of the action given the social goal and belief

Determine if the statement is likely, and respond with only either A or B. If it's not certain but it's possible, it's likely.

{inf_agent}'s social goal: {goal}
{inf_agent}'s belief: {belief}
Here is a statement of {inf_agent}'s action. Think about {inf_agent}'s goal.
{inf_agent} will perform actions according to {inf_agent}'s belief, and any action that does not align with the belief is very unlikely, except when {inf_agent}'s goal is to hinder or to prevent others. In this case (goal is hindering others) {inf_agent}'s action is only likely when it's different from {inf_agent}'s belief. If {inf_agent}'s mental states contain conditions like "When giving information" and the action is not giving information, it's unlikely.

Determine if the following statement is likely: {action}
A) Likely.
B) Unlikely.

### Estimating the likelihood of the action given the goal and belief

Determine if the statement is likely, and respond with only either A or B. If it's not certain but it's possible, it's likely.
{inf_agent}'s social goal: {goal}
{inf_agent}'s belief: {belief}
Here is a statement of {inf_agent}'s action. The belief stands for {inf_agent}'s current belief. {inf_agent} is likely to act according to goal and belief concerning certain objects (the wording for objects must be same. You should ignore the correlation of different objects. e.g., plate and apple are two different objects.) Notice that {inf_agent}'s belief does not represent the goal.
When belief and goal are irrelevant, and action is driven by goal, it's likely. When belief and goal are relevant (about exactly the same object) and they contradict with action, it's unlikely.

Determine if the following statement is likely: {action}
A) Likely.
B) Unlikely.

### Estimating the likelihood of the best action among choices given the goal and belief

Determine if the statement is likely, and respond with only either A or B. If it's not certain but it's possible, it's likely.
{inf_agent}'s belief: {belief}
{inf_agent}'s goal: {goal}
If the next immediate actions possible are: {actions}
Determine which immediate action is most possible given the information about {inf_agent}'s goal and belief.

Determine if the following statement is likely: {action_a} is a better immediate action than {action_b}.
A) Likely.
B) Unlikely.

### Estimating the likelihood of the initial belief

Determine if the statement is likely, and respond with only either A or B. If it's not certain but it's possible, it's considered likely.
Here is a statement of the story and {inf_agent}' initial belief.
There is an action that causes the state of the main object to change. Based on {inf_agent}'s observations determine if {inf_agent} perceives the state of the object change.
If it is not clearly stated that {inf_agent} perceives it then we do not assume that {inf_agent} perceived the change of state.

If {inf_agent} perceives this change then it is highly likely that {inf_agent}'s belief aligns with the change of state of the object.

If {inf_agent} does not perceive this change or if it is unknown if {inf_agent} perceives this change then it is highly likely that {inf_agent}'s belief does not align with the change of state of the object.

Story: {story}

Think about the state of the world and others actions. {inf_agent}' belief can change throughout time through other's actions and what {inf_agent} can observe. It is also important to think about if {inf_agent} can observe other's actions. If {inf_agent} can observe the same then their belief will change and if not then their belief will remain constant. Use this to determine {inf_agent}'s beliefs.

Determine if the following statement is likely: {statement}
A) Likely.
B) Unlikely.

---

### Estimating the likelihood of the belief given the observation and previous belief

Determine if the statement is likely, respond with only either A or B.
{inf_agent}'s previous belief: {previous_belief}
{inf_agent}'s observation: {observation}
Here is a statement of {inf_agent}'s current belief. If {inf_agent}'s current belief is not aligned with {inf_agent}'s observation, it is very unlikely.

Determine if the following statement is likely: {statement}
A) Likely.
B) Unlikely.

---

### Estimating the likelihood of the belief given the state and previous belief

Determine if the statement is likely, respond with only either A or B.
{inf_agent}'s previous belief: {belief}
State: {state}
Here is a statement of {inf_agent}'s current belief. If {inf_agent}'s current belief is not aligned with the state, it is very unlikely.

Determine if the following statement is likely: {statement}
A) Likely.
B) Unlikely.

---

### Estimating the likelihood of the utterance

Determine if {inf_agent}'s utterance is likely, and respond with only either A or B.
{inf_agent}'s belief: {belief}
{inf_agent}'s goal: {goal}
{inf_agent}'s utterance: {utterance}
When {inf_agent}'s goal is to help others, {inf_agent}'s utterance is likely when it strictly reflects {inf_agent}'s belief, and unlikely if it does not reflect {inf_agent}'s belief.
When {inf_agent}'s goal is to hinder or to prevent others from achieving their goals, {inf_agent}'s utterance is likely when it's different from {inf_agent}'s belief, and unlikely if it reflects {inf_agent}'s belief.

Determine if {inf_agent}'s utterance is likely.
A) Likely.
B) Unlikely.

## F.4 Initial Model Proposal

We use the following prompts to propose an initial model for a question and determine if the question has higher-order beliefs.

---

**Proposing the initial model**

What variables are necessary to solve this question? Please provide the answer without an explanation.
Please select from the following: ["State", "Observation", "Belief", "Action", "Goal"]
State: The true condition of the environment. This should always be included.
Observation: The observed information about the state. Include this when the agent has partial observations of the state.
Belief: The agent's current estimation of the true state is based on the state or its observation.
Action: A move made by the agent, informed by the state or belief. Include this only when it is directly relevant to answering the question.
Goal: The objective the agent is trying to achieve. Include this only if "Action" is included.

Question:{example_question}
Variables: {example_answer}
Question: {question}
Variables:

---

**Determining if the question contains a higher-order belief**

Determine whether the question is about a higher-order belief.
A higher-order belief refers to a belief about another person's belief, goal, or action.
It is not a high-order belief if it only asks about one agent's belief.
Please respond with "Yes" or "No".
If the answer is "Yes", the question often ends with "Where does A think that B ...?" Otherwise, respond "No".

Question: [A story involving several people.] Where will Jack look for the celery?
Higher-order belief: No
Question: [A story involving several people.] Where does Jack think that Chloe searches for the hat?
Higher-order belief: Yes
Question: {question}
Higher-order belief:

---

