# OpenReview forum: "AutoToM: Scaling Model-based Mental Inference via Automated Agent Modeling"
_NeurIPS.cc/2025/Conference — NeurIPS 2025 spotlight_

### Official Review · Reviewer_B8dt · 2025-06-29

**Clarity:** 3
**Significance:** 4
**Originality:** 4
**Rating:** 5
**Confidence:** 4

**Summary:**

This paper contributes an automated method for combining LLMs and Bayesian Inverse Planning (BIP) for modeling an agent’s Theory of Mind (ToM). This method works by iterating between agent model discovery (proposing a model, extracting the values of observable variables, and potentially adding variables or timesteps) and BIP (proposing hypotheses for each latent mental variable and evaluating the likelihood of the query given the observations). The authors show this method is capable of outperforming other LLM-based ToM methods across a number of ToM benchmarks.

**Questions:**

- Given the definition of the model utility in Eqn. (5), how does AutoToM avoid getting into local optima of models that are both overly simple and certain about the wrong answers to the queries?
- How does the performance of AutoToM vary when using different base LLMs?

**Ethical Concerns:**

["NO or VERY MINOR ethics concerns only"]

**Final Justification:**

The authors have clearly addressed the my questions/concerns with further results.

**Limitations:**

Yes

**Quality:**

3

**Strengths And Weaknesses:**

Strengths:
- Each aspect of AutoToM is well-motivated, explained clearly, and well-evaluated with the ablation studies.
- The quality and significance of this contribution is well supported by the breadth of benchmarks and baselines that are considered.

Weaknesses:
- The benchmark performance results in Table 1 and Figure 4 do not contain any error bars. The improved performance of AutoToM could be better established using multiple runs for each test case.

---

> ### Author Rebuttal · Authors · 2025-07-31
>
> Thank you for your insightful review and for recognizing AutoToM’s clear motivation and explanations, strong ablation studies, and the wide set of benchmarks and baselines that support its quality and significance. We appreciate your constructive feedback and will address your concerns as follows.
>
> > **W1. The benchmark performance results in Table 1 and Figure 4 do not contain any error bars. The improved performance of AutoToM could be better established using multiple runs for each test case.**
>
> Due to budget constraints, we were unable to run every benchmark multiple times. However, for the most challenging benchmark in terms of baseline performance, MMToM-QA, we evaluated AutoToM using three different random seeds. The average performance had a mean accuracy of 82.56% with a standard error of 0.45%. This is consistent with the accuracy of 83.00% reported in Table 1 and indicates relatively stable performance across different runs. Similarly, we also ran o3-mini-high with three different random seeds. It had a mean accuracy of 65.94% with a standard error of 0.59%. This result shows that the evaluation is quite stable and our findings still hold after multiple runs.
>
> > **Q1. Given the definition of the model utility in Eqn. (5), how does AutoToM avoid getting into local optima of models that are both overly simple and certain about the wrong answers to the queries?**
>
> This is indeed an insightful question.
>
> 1. We do not begin with an overly simple model—we start with an initial model proposal. For example, when addressing a question about belief, we start with models that incorporate beliefs. Utterances and actions are also explicitly considered during the initial model proposal. In practice, this initial model is usually rich enough and requires only minor adjustments over 1 or 2 iterations.
>
> 2. For AutoToM to be confident, the model needs to produce high action/utterance likelihood, P(a | relevant variables) or P(u | relevant variables), to fully account for the observed behavior. It’s difficult for the model to produce high action/utterance likelihood if crucial variables are missing. E.g., if the agent has only partial observations but belief is missing in the agent model, the model would not explain its behavior caused by perishable observability.
>
> 3. The only exception we observed in the experiments is that there were very few instances (2 out of 600 questions) in MMToM-QA, where a simpler model (MDP) is used when a more complex model (POMDP) would be more appropriate. This is caused by errors in the GPT-4o’s estimation of action likelihood. As a potential solution, AutoToM supports improving the model using human feedback (see Appendix B). We can also use MCMC for model adjustment to better escape local optima.
>
> > **Q2. How does the performance of AutoToM vary when using different base LLMs?**
>
> To test AutoToM’s performance sensitivity to LLM backends, we conducted additional experiments using alternative models. Note that we used the **same prompt** for each backend LLM. Specifically, we replace the GPT-4o backend with Qwen3-235B (open-sourced), DeepSeek-V3 (open-sourced), and Gemini-2.5-flash (thinking disabled) on the most challenging MMToM-QA benchmark. Notably, AutoToM with any LLM as the backend outperforms the corresponding LLM performance by a large margin (Rebuttal Table 1). Crucially, we achieve this without extra prompt engineering.
>
> Rebuttal Table 1: Model performance comparison on MMToM-QA. LLM indicates the model itself; AutoToM represents our method with corresponding model as the backend.
> | | LLM | AutoToM |
> | :--- | :--- | :--- |
> | GPT-4o | 44.0 | 83.0 |
> | Qwen3-235b-a22b-2507 | 45.0 | 67.5 |
> | DeepSeek-chat-v3-0324 | 34.8 | 71.1 |
> | Gemini-2.5-Flash (thinking disabled) | 44.7 | 71.7 |

---

> > ### Comment · Reviewer_B8dt · 2025-08-02
> >
> > Thank you for responding in depth to all of my questions/concerns. Please add these additional results into the paper, space permitting.

---

> ### Author Response · Authors · 2025-08-06
>
> Thank you for your positive feedback — we're glad our clarifications were helpful. We truly appreciate the time and effort you've devoted to reviewing our work. We will incorporate the additional results into a future revision.

---

### Official Review · Reviewer_9KJy · 2025-06-30

**Clarity:** 3
**Significance:** 3
**Originality:** 3
**Rating:** 5
**Confidence:** 2

**Summary:**

This paper introduces AutoToM, a framework for automating agent model construction and Bayesian inverse planning in Theory of Mind (ToM) inference tasks. Leveraging a large language model (LLM) as a computational backend, AutoToM dynamically selects relevant mental variables and timesteps from context to construct a suitable agent model. It performs iterative model refinement guided by uncertainty-based utility metrics. Experimental results across five diverse ToM benchmarks, cognitive studies, and embodied reasoning tasks show that AutoToM consistently outperforms state-of-the-art ToM methods, including prompting-based and model-based approaches.

**Questions:**

Cost Function Clarification:
Does the model complexity cost C(M) include both variable and timestep dimensions? Clarifying this could help readers understand how model parsimony is enforced.

Hi-ToM Performance:
In Table 1 and Figure 4d, AutoToM appears to underperform on Hi-ToM and high-order mind recursion. What challenges does the framework face in higher-order recursive tasks, and how does it decide when to expand to a higher-order agent model during agent-model automation?

LLM Backend Sensitivity:
Given that AutoToM relies on the capabilities of the underlying LLM, it would be helpful to evaluate how the quality of proposed agent models and final inference performance vary across different LLM backends.

**Ethical Concerns:**

["NO or VERY MINOR ethics concerns only"]

**Final Justification:**

Thanks for the rebuttal. It addresses my questions. I raised my ratings accordingly.

**Quality:**

3

**Strengths And Weaknesses:**

Strengths:
1. AutoToM addresses a key limitation of existing model-based ToM methods by eliminating manual model specification. Its automated model discovery enables generalization to open-domain and higher-order reasoning tasks.
2. The framework dynamically selects the appropriate set of mental variables (e.g., goal, belief, observation) and relevant timesteps, making it adaptable to different ToM problems. Figure 2a effectively illustrates the variety of agent models AutoToM can construct.
3. The paper presents thorough evaluations from multiple perspectives—including accuracy, context length, number of agents, and recursion depth. Qualitative case studies in Sections 4.2 (cognitive plausibility) and 4.3 (embodied assistance) further highlight the practical utility and interpretability of AutoToM.

Weaknesses:
1. Since AutoToM performs agent model discovery per query, it may incur higher inference latency compared to methods with static models. A comparison of inference time across methods would help contextualize this trade-off.
2. The paper briefly mentions variable and timestep adjustments are done iteratively during model refinement, but a more detailed explanation or visualization of how the system alternates between these adjustments would improve transparency and reproducibility.

---

> ### Author Rebuttal · Authors · 2025-07-31
>
> Thank you for your insightful review and for recognizing that AutoToM removes manual model specification with automated model discovery, generalizes to open‑domain and higher‑order reasoning, and is backed by thorough evaluations and case studies showing practical utility and interpretability. We appreciate your constructive feedback and will address your concerns as follows.
>
> > **W1. Since AutoToM performs agent model discovery per query, it may incur higher inference latency compared to methods with static models. A comparison of inference time across methods would help contextualize this trade-off.**
>
> While AutoToM outperforms large reasoning models in terms of performance, it is also faster or comparable in real-world scenarios. For embodied tasks, AutoToM takes 30 seconds per timestep, which is only slightly slower than its backend LLM, GPT-4o (20 seconds per timestep). o3-mini, by contrast, requires 100 seconds per timestep, making it impractical for embodied decision-making. Additionally, we analyzed the inference speed and computational cost of AutoToM and reasoning models on the most computationally demanding ToM benchmark, MMToM-QA (due to long context). The comparisons, as shown in Rebuttal Table 1, show that AutoToM requires fewer or comparable tokens and inference time.
>
> Rebuttal Table 1: Token cost and inference time comparison on MMToM-QA.
> | | Average #Tokens per Question (thousands) | Average Inference Time (seconds) |
> | :--- | :--- | :--- |
> | AutoToM | 8.0 | 8.5 |
> | o3-mini-high | 10.9 | 21.6 |
> | Gemini 2.0 Flash Thinking | 8.8 | 6.1 |
>
> > **W2. A more detailed explanation or visualization of how the system alternates between variable and timestep adjustments would improve transparency and reproducibility.**
>
> Thank you for the constructive feedback. In the paper, *Figure 1* overviews the AutoToM procedure; *Figure 3* illustrates example hypotheses and calculations for Bayesian inverse planning; and *Figure 5* shows a real example of AutoToM’s model adjustment and inference in a BigToM false-belief scenario. *Appendix C.3* presents two real examples demonstrating variable adjustment (Example 1) and timestep adjustment (Example 2) in AutoToM.
>
> Following your suggestion, we will add an additional example figure that clearly presents every element (each hypothesis, each calculation) and every step of the process (initial model → inference results → variable adjustment → inference results → timestep adjustment → …), with easily digestible annotations.
>
> > **Q1. Cost Function Clarification**
>
> Thanks for this question. No, our complexity cost is C(M)=α∣M∣, where ∣M∣ counts only the number of latent mental variable types; it does not include timesteps. We start from the last timestep and only add earlier steps if the utility remains low, so parsimony is enforced by this conservative timestep expansion, and with the cost term.
>
> > **Q2. Hi-ToM Performance**
>
> **Challenges with higher-order tasks.** For higher-order tasks, AutoToM samples one possible state based on b(s) at level *l* to approximate the state at level *l* - 1 as imagined by the agent at level *l*. This process is applied recursively down to level 0, where a standard automated BIP is performed.   However, **as recursion deepens, errors may accumulate**: inaccuracies in belief estimation at any level or state can propagate downward, potentially affecting reasoning at level 0. To further improve the accuracy of recursive reasoning, we need to reduce errors introduced during BIP when sampling beliefs—specifically, by enhancing the accuracy of non-recursive belief inferences.
>
> Other errors occur when AutoToM fails to recognize the relevance of certain mental variables, resulting in an insufficient model. This may be improved with better LLM backends, and we will also explore broader improvements at the framework level, such as learning-based approximations to replace the computationally expensive nested BIPs.
>
> **Deciding on higher-order inference.** The initial model proposal assesses the required level of recursive reasoning for higher-order ToM inference. This level is more fixed and easier to determine early on compared to the necessary variables and timesteps, which we adjust dynamically based on inference uncertainty.
>
> > **Q3. LLM Backend Sensitivity**
>
> To test AutoToM’s performance sensitivity to LLM backends, we conducted additional experiments using alternative models. Note that we used the **same prompt** for each backend LLM. Specifically, we replace the GPT-4o backend with Qwen3-235B (open-sourced), DeepSeek-V3 (open-sourced), and Gemini-2.5-flash (thinking disabled) on the most challenging MMToM-QA benchmark. Notably, AutoToM with any LLM as the backend outperforms the corresponding LLM performance by a large margin (Rebuttal Table 2). Crucially, we achieve this without extra prompt engineering.
>
> Rebuttal Table 2: Model performance comparison on MMToM-QA. LLM indicates the model itself; AutoToM represents our method with corresponding model as the backend.
> | | LLM | AutoToM |
> | :--- | :--- | :--- |
> | GPT-4o | 44.0 | 83.0 |
> | Qwen3-235b-a22b-2507 | 45.0 | 67.5 |
> | DeepSeek-chat-v3-0324 | 34.8 | 71.1 |
> | Gemini-2.5-Flash (thinking disabled) | 44.7 | 71.7 |

---

### Official Review · Reviewer_Cfs5 · 2025-07-03

**Clarity:** 3
**Significance:** 3
**Originality:** 3
**Rating:** 5
**Confidence:** 2

**Summary:**

This paper introduces AutoToM, a framework for scalable Theory of Mind (ToM) reasoning that automatically constructs agent models and performs Bayesian inverse planning using LLMs. Unlike prior work that relies on handcrafted model structures, AutoToM discovers relevant variables (e.g., beliefs, goals) and their values dynamically through LLM prompting. The system is evaluated across ToM benchmarks and cognitive studies, showing good performance in both accuracy and human-aligned confidence estimation.

**Questions:**

To what extent does AutoToM's performance depend on the specific LLM used? Could smaller models offer similar benefits with prompt engineering?

Can the system be extended to jointly learn both variable structure and causal dependencies, rather than fixing the structure a priori?

**Ethical Concerns:**

["NO or VERY MINOR ethics concerns only"]

**Final Justification:**

The rebuttal provided clear and detailed clarifications that addressed my main concerns. In particular, it explained the flexibility of the model structure within principled constraints, the focused role of LLMs limited to local conditional estimation, and the use of probabilistic inference for the remainder of the pipeline

**Limitations:**

AutoToM’s inference relies heavily on LLM-generated hypotheses and likelihood approximations, which may introduce bias or inconsistencies.

The high computational cost (average 27 API calls per question) poses practical limitations for real-time or large-scale deployment.

**Quality:**

3

**Strengths And Weaknesses:**

Strengths:
- The paper proposes a novel integration of LLM-driven model structure discovery with Bayesian inverse planning.
- AutoToM achieves state-of-the-art performance across multiple ToM benchmarks with particularly strong gains on classic false-belief tasks.
- The framework generalizes across diverse tasks and domains, and demonstrates human-like confidence calibration.

Weaknesses:
- While AutoToM claims to automate model discovery, the underlying causal structure is fixed, limiting the scope of structural generalization.
- The framework does not learn the variables or their dependencies; instead, it selects and scores hypotheses using LLM outputs without probabilistic grounding.
- Some of the evaluation gains may stem from prompt-level tuning or LLM scoring behavior, rather than principled modeling.

---

> ### Author Rebuttal · Authors · 2025-07-31
>
> Thank you for your insightful review and for recognizing AutoToM’s novelty, state‑of‑the‑art results, and its generalization with human‑like confidence calibration. We appreciate your constructive feedback and will address your concerns as follows.
>
> > **W1 & Q2. While AutoToM claims to automate model discovery, the underlying causal structure is fixed, limiting the scope of structural generalization. Can the system be extended to jointly learn both variable structure and causal dependencies, rather than fixing the structure a priori?**
>
> AutoToM jointly learns both variable structure and causal dependencies, but only within a space of plausible causal relationships. It automates agent model discovery by identifying the necessary mental variables and timesteps while preserving causal dependencies within a family of structures grounded in standard decision-theoretic primitives (state, observation, belief, action, goal). For example, in partially observable scenarios, belief depends on observation, which in turn depends on state; in fully observable scenarios, AutoToM simplifies this by having belief depend directly on state.
>
> These causal dependencies are constrained to align with causal structures established in prior agent modeling works (e.g., Kaelbling et al., 1998; Gmytrasiewicz and Doshi, 2005) and cognitive studies (e.g., Baker et al., 2009; Baker et al., 2017; Ullman et al., 2009). For example, actions may depend on observations or beliefs, and optionally on goals, but observations **cannot** depend on the agent’s own actions, since observations reflect what the agent perceives, regardless of its own actions. Rather than introducing implausible causal dependencies, we selectively include and dynamically adjust dependencies within the model space. While extending beyond classical MDP/POMDP/I‑POMDP formulations, AutoToM's model space supports 30 configurations per timestep and allows for arbitrary levels of recursive reasoning, as discussed in Appendix A.5.
>
> Empirically, AutoToM’s model structure generalizes across domains and modalities: we evaluated it on five ToM benchmarks, two cognitive studies, and an embodied assistance task, demonstrating strong overall results and stable behavior across diverse contexts, number of agents, modalities, and recursion level.
>
> To further enhance generalization, domain-specific hypothesis libraries or other inductive biases for proposing variables can be incorporated, without changing the core algorithm. For instance, to enable affective reasoning—specifically, attitude inference as in the OpenToM benchmark—we extended the set of latent variables to include preference and attitude, and constructed new causal dependencies accordingly. This modification allowed AutoToM to perform affective inference effectively, achieving a significantly higher Macro-F1 score than GPT-4o (0.56 vs. 0.48).
>
> > **W2. The framework does not learn the variables or their dependencies; instead, it selects and scores hypotheses using LLM outputs without probabilistic grounding.**
>
> The hypotheses are scored and selected using probabilistic inference rather than by LLM outputs alone. This selection process is illustrated in the key steps in Figure 3: Step 1 is formulated by our model and estimated with the help of an LLM, while Steps 2 and 3 do not involve the LLM.
>
> **Step 1:** Estimating local conditionals: $P(b_1^t  | b_1^{t-1}, o_2^t)=0.4$
>
> In this step, the local conditional is estimated by an LLM. Given the clear dependencies, it's significantly more accurate for an LLM to estimate this than to tackle the entire problem directly.
>
> **Step 2:** Calculating posterior probabilities: $P(v_i^t| X^{t_{s}:t}) \propto \sum_{V_{-i}^{t_s:t}} P(v_i^t, V_{-i}^{t_{s}:t}, X^{t_{s}:t})$
>
> This step relies purely on probabilistic inference to explicitly compute the final probability distribution, without using the LLM.
>
> **Step 3:** Model adjustment based on utility: $U(M, q) = R(M, q) - C(M)$
>
> This step involves model adjustment based on the computed utility, which accounts for both the model’s confidence and cost. The LLM is not used in this step.
>
> > **W3. Some of the evaluation gains may stem from prompt-level tuning or LLM scoring behavior, rather than principled modeling.**
>
> The success of AutoToM mainly comes from constructing agent models (Bayesian networks) and conducting Bayesian inference based on these constructed agent models. It substantially outperforms all previous prompting-based methods, such as SymbolicToM and SimToM, particularly in long-context and complex scenarios. These gains result from a principled agent modeling approach, decomposing the problem and clearly modeling mental dependencies.
>
> Concretely, AutoToM extracts observable variables, constructs explicit mental models, and performs probabilistic inference using Bayesian inverse planning, grounded in cognitive studies. The agent models it constructs align with those commonly used by cognitive scientists. AutoToM iteratively refines these models to maximize confidence while minimizing complexity. The model adjustment is guided by a formal objective defined by our model utility (negative entropy of posterior distribution as reward and model complexity as cost), rather than relying on LLM outputs alone.
>
> > **Q1. To what extent does AutoToM's performance depend on the specific LLM used? Could smaller models offer similar benefits with prompt engineering?**
>
> To test AutoToM’s performance sensitivity to LLM backends, we conducted additional experiments using alternative models. Note that we used the **same prompt** for each backend LLM. Specifically, we replace the GPT-4o backend with Qwen3-235B (open-sourced), DeepSeek-V3 (open-sourced), and Gemini-2.5-flash (thinking disabled) on the most challenging MMToM-QA benchmark. Notably, AutoToM with any LLM as the backend outperforms the corresponding LLM performance by a large margin (Rebuttal Table 1). Crucially, we achieve this without extra prompt engineering.
>
>
> Rebuttal Table 1: Model performance comparison on MMToM-QA. LLM indicates the model itself; AutoToM represents our method with corresponding model as the backend.
> | | LLM | AutoToM |
> | :--- | :--- | :--- |
> | GPT-4o | 44.0 | 83.0 |
> | Qwen3-235b-a22b-2507 | 45.0 | 67.5 |
> | DeepSeek-chat-v3-0324 | 34.8 | 71.1 |
> | Gemini-2.5-Flash (thinking disabled) | 44.7 | 71.7 |

---

> > ### Comment · Reviewer_Cfs5 · 2025-08-09
> >
> > The rebuttal provided clear and thorough responses to the main concerns. In particular, it clarified the flexibility of model structure within principled constraints, explained the limited but focused role of LLMs in the pipeline, and supplied additional experiments showing robustness across multiple LLM backends. These points, along with concrete examples of domain extension and performance gains beyond prompt-based baselines, substantially strengthened the case for the work.

---

> > > ### Author Response · Authors · 2025-08-09
> > > **Thanks for your feedback**
> > >
> > > Dear reviewer,
> > >
> > > Thank you for your valuable comments and for your positive assessment of our rebuttal. We are glad that our clarifications and additional experiments addressed your concerns and helped strengthen the paper. We truly appreciate the time and effort you've devoted to reviewing our work.

---

### Official Review · Reviewer_J3da · 2025-07-07

**Clarity:** 2
**Significance:** 3
**Originality:** 4
**Rating:** 5
**Confidence:** 4

**Summary:**

The paper introduces a method for modeling agents and enable interpretable mental inference called AutoToM. AutoToM proposes and iteratively refines an agent model through Bayesian inference planning, leveraging an LLM for every component. AutoToM outperforms out-of-the-box LLMs and LRMs plus other existing ToM methods on five well-known ToM (often false-belief) benchmarks. Interestingly, AutoToM produces confidence estimates that are much more aligned with humans and shows great applicability in tasks such as embodied decision-making.

**Questions:**

(besides the ones in the "weaknesses" items above)

- In Figure 4, do you have any insights as of to what cases does AutoToM excels at, and what are issues still to tackle to take e.g. "Level 2+ Belief" from 70% -> 100%?
- Would AutoToM also improve performance for benchmarks that sometimes require affective ToM such as OpenToM?
- Why do you mention that AutoToM is more *robust*? Specifically, since AutoToM is LLM-based, I assume it'll still be prone to issues such as those reported in ExploreToM (e.g. slightly different story structures triggering ToM issues due to LLMs' flakiness). It would be important to specify exactly robust to what behavior (since increased performance != robustness), or maybe considering not focusing on that claim given the high value in other aspects.

I believe this is overall a strong paper, but I would really appreciate resolving these remaining concerns.

**Ethical Concerns:**

["NO or VERY MINOR ethics concerns only"]

**Final Justification:**

Raising my score from 4 to 5 based on the extra detailed explanations, comparisons with prior work, and extra experiments. It's particularly great to hear that AutoToM is efficient w.r.t. reasoning models!

**Limitations:**

yes

**Paper Formatting Concerns:**

-

**Quality:**

3

**Strengths And Weaknesses:**

Strengths:
- The paper tackles an exciting and important problem: modeling theory of mind without overly imposing structure and thus reducing the applicability of the method, as is often the case with many current methods. The paper demonstrates that AutoToM achieves this by simultaneously exhibiting strong performance on ToM benchmarks, while also evaluating beyond ToM benchmarks (embodied decision-making; human uncertainty comparison).
- Generally strong set of experiments! Some remaining concerns below.

Weaknesses:
- The paper could be written more clearly, with for example an example figure that is more easily digestible and representative of the procedure.
- Some literature should be discussed more deeply, particularly thought-tracing [22] since it shares the same claim of wide-applicability in ToM modeling using a Bayesian approach, and yet it is clumped with more structured methods like SymbolicToM [34]. It would be beneficial to clearly discuss the differences with AutoToM and benefits of the present approach! Moreover, it would be important to include other types of approaches in related work, such as training LLMs specifically for ToM, as shown by e.g. ExploreToM (Sclar et al., 2024) or Textual Time Travel (Arodi and Cheung, 2021).
- Would be good to add a token cost vs performance discussion: this is already included in an ablation, but it would be good to know the cost of AutoToM versus other methods.
- Some experimental concerns: comparing AutoToM with thought-tracing, I see some very different baseline accuracies being reported, such as MMToM-QA GPT-4o (44.00 in AutoToM; 56.5 in thought-tracing) and o3-mini. Why are there some big differences? Is it purely a model version issue? Also, would it be possible to add FANToM, since it disguises more the false-belief task than ToMi/Hi-ToM, and according to thought-tracing it seems to be the most challenging?

---

> ### Author Rebuttal · Authors · 2025-07-31
>
> Thank you for your insightful review and for recognizing that our paper addresses an important challenge and presents strong experiments. We will address your concerns as follows.
>
> > **W1. The paper could be written more clearly, with a more digestible and representative example figure.**
>
> Thank you for the constructive feedback. In the paper, *Figure 1* overviews the AutoToM procedure; *Figure 3* illustrates example hypotheses and calculations for Bayesian inverse planning; and *Figure 5* shows a real example of AutoToM’s model adjustment and inference in a BigToM false-belief scenario. *Appendix C.3* presents two real examples demonstrating variable adjustment (Example 1) and timestep adjustment (Example 2) in AutoToM.
>
> Following your suggestion, we will add an additional example figure that clearly presents every element (each hypothesis, each calculation) and every step of the process (initial model → inference results → variable adjustment → inference results → timestep adjustment → …), with easily digestible annotations.
>
> > **W2. Some literature should be discussed more deeply.**
>
> Thank you for the suggestion. We will include more discussion on thought-tracing and other related works. We clarify how AutoToM differs below.
>
> **Key difference 1: explicit agent modeling**. *AutoToM* constructs agent models (Bayesian networks) and conducts Bayesian inference based on them, specifying relevant mental variables and their causal relations. They describe the decision-making processes that lead to the observed agent behaviors. In contrast, *thought-tracing* maintains hypotheses only about the mental variable asked in the question, without the structural representations of other related mental variables. This explicit modeling yields higher accuracy on hard benchmarks and robustness to wording or superficial story changes (e.g., thought-tracing needs wording changes for MMToM-QA, while AutoToM does not). Moreover, error sources are easier to identify in AutoToM by inspecting its model structure and variables.
>
> **Key difference 2: inference complexity minimization**. *AutoToM* discovers a minimally sufficient model per query: it proposes an initial model and adds variables and/or timesteps only when they improve utility. This avoids under-/over-modeling and scales with longer contexts, more agents, and deeper recursion. As shown in Exp. 3, this complexity control makes AutoToM much more efficient than reasoning models on downstream embodied assistance tasks. By contrast, *thought-tracing* only reweights hypotheses and does not adapt hypothesis complexity or timesteps.
>
> > **W3. Would be good to add a token cost vs performance discussion.**
>
> Following your suggestion, we added a cost analysis of AutoToM vs. other methods on MMToM-QA, the most challenging dataset due to long context. The average number of tokens per question for AutoToM, o3-mini-high, and Gemini 2.0 Flash Thinking are 8.0k, 10.9k, and 8.8k, respectively. This shows that AutoToM outperforms large reasoning models while using fewer tokens.
>
> > **Q1. Difference in reported baseline accuracies; adding FANToM.**
>
> The discrepancy in baseline performance on MMToM-QA is caused by:
>
> 1. Dataset modifications: The authors replaced object and room names with symbols like A, B, etc.
>
> 2. Modalities: The authors used a text-only version, while we use translated multimodal fusion. The text-only version is significantly easier—with the same model, o3-mini achieves 71.5% accuracy on text-only versus 64.7% on the multimodal version.
>
> 3. Number of questions: The authors evaluated on a subset of 200 questions, whereas we tested on the full benchmark of 600 questions.
>
> Similarly, thought-tracing also used subsets for other benchmarks. Additionally, they relabeled the answers in BigToM.
>
> **FANToM:** To further demonstrate AutoToM’s ability to solve false-belief tasks, we tested AutoToM on FANToM. We randomly selected a subset of 200 false-belief first-order questions with short contexts due to time and budget constraints.
>
> **Results.** AutoToM, with a GPT-4o backend, achieved 72.7%, outperforming the GPT-4o baseline, which achieved 57.5%. AutoToM, with a Gemini 2.5 Flash backend, achieved 77.9%, outperforming the Gemini 2.5 Flash baseline, which achieved 38%. With either model as the backend LLM, AutoToM improves upon the original baselines.
>
> **Analysis.** AutoToM is able to solve false belief questions by extracting the essential variables. In FANToM, AutoToM extracts the state of the conversation (the agents in the conversation, if the main agent is currently in the conversation, and the topics discussed), utterances, and observation of the main agent (depending on whether they are in the conversation or not) to infer belief. In contrast, the two baselines struggle to accurately extract and track the agent’s observation throughout the conversation.
>
> > **Q2. In Figure 4, what cases does AutoToM excel in, and what challenges remain to improve "Level 2+ Belief" from 70% to 100%?**
>
> **Where AutoToM excels**. AutoToM is notably stable and scales effectively as contexts grow longer and more agents are introduced, outperforming large reasoning models in robustness. This is due to its explicit construction and adjustment of a causal agent model, followed by Bayesian inverse planning over latent mental variables, rather than relying solely on prompt heuristics.
>
> AutoToM also performs better at goal inference, which all LLMs and large reasoning models struggle with. This is because AutoToM conducts explicit inverse planning, which reasons about how agents act conditioned on certain goal hypotheses and other relevant mental variables (e.g., beliefs). In contrast, LLMs and large reasoning models often produce incorrect goal inferences due to spurious action patterns, particularly when there is partial observability, where action plans are often not obvious due to incomplete information or false beliefs.
>
> **How to push toward 100%**. Errors mainly arise from noisy multimodal extraction, imprecise probability estimates, and occasionally missing mental variables. For Level-2+ Belief, recursive belief sampling (from level *l* down to 0) **accumulates errors as recursion deepens**. Improving base (non-recursive) belief inference and BIP estimation will reduce errors. Other errors occur when AutoToM misses relevant mental variables, producing an insufficient model. We will mitigate this with stronger LLM backends and framework-level advances, including learning-based approximations to replace costly nested BIPs.
>
> > **Q3. Would AutoToM also improve performance for benchmarks that sometimes require affective ToM such as OpenToM?**
>
> Thanks for the insightful question. We evaluated AutoToM’s affective ToM by extending the causal structure to include *attitude* and *preference* (all other components unchanged) and testing on all 596 OpenToM attitude questions.
>
> **Results.** Following the OpenToM paper (Xu et al., 2024), we used Macro-F1 as the evaluation metric. The random baseline is 0.33. GPT-4o achieved 0.48, while AutoToM with GPT-4o backend outperformed it with a score of 0.56. AutoToM also approached the performance of the large reasoning model o3-mini-high (0.60), indicating its strong affective reasoning capability.
>
> **Analysis.** Answering the attitude questions does not require inverse planning, since the model can just directly perform forward estimation of attitude based on observed events and preference. This explains why AutoToM performed similarly compared to o3-mini-high. This is consistent with results for other question types that do not require inverse planning, such as level 0 (no ToM) and level 1 action questions shown in Figure 4a. However, even in the case where inverse planning is not required, AutoToM still scores higher than its backend LLM (GPT-4o). We attribute this to AutoToM’s ability to extract and focus on variables that are causally relevant to the task, while filtering out spurious cues by design (see Xu et al., 2024, Section 2.5) that may mislead GPT-4o.
>
> > **Q4. Why describe AutoToM as more robust? It would be important to specify exactly robust to what behavior.**
>
> Thanks for the suggestion. We define robustness as:
>
> 1. **Consistent performance under input/task variations.** AutoToM shows markedly lower variance than large reasoning models across different input/task variants, including (i) context length, (ii) number of agents, (iii) recursion level, and (iv) question type, as shown in Fig. 4. AutoToM is also more resilient to wording styles. E.g., thought-tracing requires rewording for MMToM-QA, whereas AutoToM needs no such modifications.
>
> 2. **Human-like uncertainty estimation.** AutoToM expresses human-aligned uncertainty in classic cognitive studies as shown in Experiment 2, unlike baselines.
>
> 3. **Mental state inference useful for downstream tasks.** AutoToM can be applied to downstream tasks such as embodied assistance. Specifically, while LLM baselines tend to propose uniform goal distributions due to distraction of long context, AutoToM concentrates probability on likely goals that are consistent with agent behavior, enabling helpful assistance.
>
> 4. **Human-in-the-loop correction.** AutoToM’s explicit agent models make reasoning interpretable and editable, enabling humans to provide feedback to correct errors when needed (Appendix B).
>
> Why this robustness emerges:
>
> 1. **Explicit modeling + BIP.** AutoToM constructs explicit mental models and performs probabilistic inference using Bayesian inverse planning, grounded in cognitive studies. This reduces sensitivity to superficial story edits that can disrupt direct LLM prompting (as observed in ExploreToM).
>
> 2. **Adaptive model selection.** Rather than relying on fixed agent models, AutoToM adjusts its model based on a utility score that balances confidence and complexity. This enables AutoToM to adapt to different problems.
>
> We will include this discussion in the revised Section 4.

---

> > ### Comment · Reviewer_J3da · 2025-08-07
> > **Thanks!**
> >
> > Dear authors,
> >
> > Thanks for the detailed explanations, comparisons with prior work, and extra experiments. It's particularly great to hear that AutoToM is efficient w.r.t. reasoning models! Please include these discussions in the camera ready :)
> >
> > I'm raising my score accordingly.

---

> ### Author Response · Authors · 2025-08-07
> **Thanks!**
>
> Dear reviewer,
>
> We sincerely appreciate your thoughtful engagement and for raising your score. We truly appreciate the time and effort you've devoted to reviewing our work. Your detailed suggestions have greatly helped us clarify our writing, strengthen our comparisons, and expand our discussion of related work, enhancing the quality of the paper :)

---

### Note · Authors · 2025-08-13

We appreciate the insightful and constructive feedback from the reviewers. We are encouraged that the reviewers recognized key strengths of AutoToM, including its novelty and significance in automating agent model discovery (Reviewer J3da, Cfs5, 9KJy, B8dt); strong performance across a comprehensive set of evaluations (Reviewer J3da, Cfs5, 9KJy, B8dt);  broad generalization and applicability across diverse domains (Reviewer J3da, Cfs5, 9KJy); alignment with human cognition (Reviewer Cfs5, 9KJy); interpretability (Reviewer J3da, 9KJy); well-motivated design (Reviewer B8dt) and effective illustrations (Reviewer 9KJy).

In response to the reviewers’ thoughtful feedback, we have conducted additional experiments and analyses to address the reviewers’ questions and concerns:
- Evaluated AutoToM on additional benchmarks, FANToM and OpenToM, which further validated AutoToM’s performance on challenging problems and demonstrated that AutoToM can operate with new causal dependencies for affective ToM. (Reviewer J3da, Cfs5)
- Added experiments and analysis on LLM backend sensitivity, which demonstrated AutoToM’s robustness. (Reviewer Cfs5, 9KJy, B8dt)
- Evaluated the inference time and token cost of AutoToM and reasoning models, which highlighted the efficiency of AutoToM in comparison to reasoning models. (Reviewer J3da, 9KJy)
- Tested models across multiple runs on MMToM-QA, which confirms the statistical reliability and stability of our results. (Reviewer B8dt)
- Expanded discussion and comparisons with thought-tracing and other related works. (Reviewer J3da)
- Discussed the robustness of AutoToM in detail. (Reviewer J3da)
- We will add a new example figure that provides a step-by-step walkthrough of AutoToM to enhance clarity. (Reviewer J3da, 9KJy)

As acknowledged by reviewers, these new results and analyses have addressed their questions and concerns. We will incorporate them in the revision to enhance the manuscript. We sincerely thank the reviewers once again for their valuable feedback.

---

### Decision · Program_Chairs · 2025-09-17

**Decision:**

Accept (spotlight)

**Comment:**

The paper introduces AutoToM, which automates agent-model construction and applies Bayesian inverse planning with an LLM backend. Reviewers generally agree that the problem studied is important, the approach is novel, and the model-based methods, and the empirical coverage is strong.
There were some concerns regarding the clarity and scope of the claims, and questions about additional analysis and results.
The rebuttal addressed these points with added FANToM and OpenToM results, backend-sensitivity studies showing consistent gains, token and time cost comparisons, multi-run results on MMToM-QA, and clearer distinctions from thought-tracing. All reviewers remain highly positive about the submission.